# FLAME♦: Factuality-Aware Alignment for Large Language Models

**Sheng-Chieh Lin**[1]*, **Luyu Gao**[2], **Barlas Oguz**[3], **Wenhan Xiong**[3],
**Jimmy Lin**[1], **Wen-tau Yih**[3], **Xilun Chen**[3]†

University of Waterloo[1], Carnegie Mellon University[2], Meta AI[3]

s269lin@uwaterloo.ca, xilun@meta.com

## Abstract

Alignment is a procedure to fine-tune pre-trained large language models (LLMs) to follow natural language instructions and serve as helpful AI assistants. We have observed, however, that the conventional alignment process fails to enhance the factual accuracy of LLMs, and often leads to the generation of more false facts (i.e., *hallucination*). In this paper, we study how to make the LLM alignment process more factual, by first identifying factors that lead to hallucination in both alignment steps: supervised fine-tuning (SFT) and reinforcement learning (RL). In particular, we find that training the LLM on new or unfamiliar knowledge can encourage hallucination. This makes SFT less factual as it trains on human-labeled data that may be novel to the LLM. Furthermore, reward functions used in standard RL often inadequately capture factuality and favor longer and more detailed responses, which inadvertently promote hallucination. Based on these observations, we propose *FactuaLity-aware AlignMEnt* (FLAME♦), comprised of *factuality-aware SFT* and *factuality-aware RL* through direct preference optimization. Experiments show that our proposed FLAME♦ guides LLMs to output more factual responses while maintaining their instruction-following capability.

## 1 Introduction

Alignment [Ouyang et al., 2022] is a procedure to make pre-trained large language models (LLMs) [Brown et al., 2020, Touvron et al., 2023] follow human instructions and serve as helpful AI assistants. Despite significant progress in general LLM alignment [Ouyang et al., 2022, Bai et al., 2022, Yuan et al., 2024], state-of-the-art aligned LLMs are still prone to generate false claims [OpenAI, 2023, Min et al., 2023]. In this work, we therefore attempt to advance the understanding of the underlying causes of LLM hallucination as well as its relation to the alignment procedure.

We consider the commonly seen alignment process consisting of two training phases: (1) supervised fine-tuning (SFT) [Sanh et al., 2022]; (2) reinforcement learning (RL) with human [RLHF, Ouyang et al., 2022, Bai et al., 2022] or automated feedback [RLAIF, Bai et al., 2023]. In our study, we find that both the SFT and RL steps in the standard alignment process may actually *encourage* LLMs to hallucinate. First, in the SFT stage, LLMs are fine-tuned with diverse instructions paired with human-created high-quality responses. While this leads to strong instruction-following capability [Ouyang et al., 2022, Köpf et al., 2023, Zhou et al., 2023, Touvron et al., 2023], our study shows that such human-labeled responses may present *new or unknown information* to the LLM. This, in turn, may inadvertently promote hallucination. Second, we find that the standard reward used in the RL stage

---

* This work is done during Sheng-Chieh's internship at Meta.
† Xilun and Sheng-Chieh contributed equally to this work.

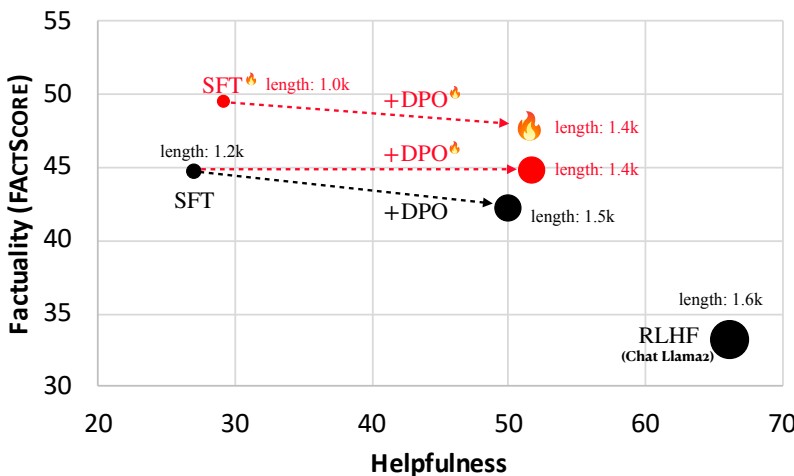

Figure 1: Models' helpfulness on Alpaca Eval vs factuality on biography. Helpfulness is measured by models' win rate over our baseline SFT + DPO on Alpaca Eval. Dot size represents average length of bio generation.

often prefers longer and more detailed responses [Singhal et al., 2023, Chen et al., 2024b, Yuan et al., 2024]. Consequently, a reward-hacking model ends up with a tendency to produce longer claims with more non-factual information, as shown in the black dots in Figure 1. One possible reason is that most existing RLHF or RLAIF approaches rely on a single scalar reward to represent preference, which struggles to cover multiple alignment skill sets [Ye et al., 2024] and is likely to under-present the aspect of factuality [Hosking et al., 2024].

To address the aforementioned issues, we study the key factors which impact factuality during alignment. In particular, we first conduct a pilot study on the biography generation task [Min et al., 2023] in a more controlled setting where the alignment process focuses solely on factuality (Section 3). Our key observation is that an LLM hallucinates more if it is fine-tuned on new knowledge in either the SFT or the RL stage. For example, an LLM becomes significantly less factual when fine-tuned on responses produced by a model with access to external knowledge (e.g. a retrieval augmented LLM), even though those responses are more factual themselves. Similarly, hallucination is greatly increased if RLAIF is performed on preference pairs that consist of retrieval-augmented LLM output as positive examples and the LLM's own output as negative examples. In comparison, we discover that fine-tuning a pre-trained LLM on a subset of its *own* generations selected by factuality yields more factual responses and reduces hallucinations.

Next, we apply our findings to improve the factuality of the general LLM alignment process, which is more challenging due to the diversity of instructions. As shown in Figure 2, we observe that some instructions require factual responses while the others do not, and therefore would require different alignment treatments. We first identify fact-based instructions that require factual responses and leverage the findings in our pilot study to create additional training data at both SFT and RL stages to explicitly guide LLMs to output factual responses. Specifically, at the SFT stage, for fact-based instructions, instead of using human created seed training data, we elicit knowledge from the pre-trained LLM and construct training data using its own pre-trained knowledge. This can prevent fine-tuning the LLM on knowledge unknown to itself. At the RL stage, we create additional preference pairs focused on factuality for fact-based instructions, which are combined with the standard preference pairs for instruction following during Direct Preference Optimization [DPO; Rafailov et al., 2023].

We evaluate models on Alpaca Eval [Dubois et al., 2024] and Biography, using win rate for instruction-following capability and FACTSCORE [Min et al., 2023] for factuality evaluation. As shown in Figure 1, using our FLAME method (SFT + DPO), a significantly higher FACTSCORE (+5.6 pts) is achieved compared to the standard alignment process (SFT + DPO), without sacrificing the LLM's instruction-following capability (51.2% win rate). Our ablation study also indicates that identifying fact-based instructions is the key to factual alignment in the general alignment setting.

## 2  Related Work

**Alignment.**  Since pre-trained LLMs cannot accurately follow human instructions, a bunch of work has been proposed to improve LLM alignment through SFT and RL. Some propose to improve SFT through data curation [Zhou et al., 2023, Chen et al., 2024a], diverse instruction augmentation [Wang et al., 2023a, Li et al., 2024] while others focus on RL with human feedback [Ouyang et al., 2022, Bai et al., 2022], AI feedback [Bai et al., 2023, Sun et al., 2024, Yuan et al., 2024]. The main goal of these alignment approaches is instruction-following capability (or helpfulness), which may guide LLMs to output detailed and lengthy responses [Singhal et al., 2023] but inevitably encourage hallucination.

**Factuality.**  Prior work has highlighted the issue of hallucination in LLMs [Gao et al., 2022, Kandpal et al., 2023, Mallen et al., 2023]. To address the issue, important research lines are factuality evaluation [Min et al., 2023, Wang et al., 2023b, Chern et al., 2023] and improvement. Some training-free approaches to improve LLMs' factuality include external knowledge augmentation [Gao et al., 2022, Kandpal et al., 2023, Cheng et al., 2023, Jiang et al., 2023] and specialized decoding [Li et al., 2023, Chuang et al., 2024].

Recent studies apply RL to improve LLMs' factuality. For example, Tian et al. [2024] propose to construct factuality preference pairs for direct preference optimization [DPO; Rafailov et al., 2023], which is closely related to our work. However, they focus solely on enhancing LLMs' factuality through DPO but overlook its potential impact on the models' instruction-following capability, as demonstrated in our experiments. In contrast, our work provides a comprehensive examination of improving LLMs' factuality and instruction-following ability through fine-tuning approaches encompassing both SFT and DPO. Concurrent to our work, Kang et al. [2024] find that LLMs tend to hallucinate when facing unfamiliar queries. They consider improving LLMs' factuality as teaching LLMs to output abstaining or less detailed responses on such unfamiliar queries, a similar behavior observed from our LLMs fine-tuned with FLAME (see case studies in Section 6.5). It is worth mentioning that both prior studies focus on a simplified scenario as our pilot study in Section 3: fine-tuning LLMs to improve factuality on a single task (e.g., fine-tuning and evaluating on biography generation). In contrast, we consider the general alignment task, where LLMs are given diverse and complex instructions.

## 3  A Pilot Study on Factual Alignment

In this section, we first study how to align large language models (LLMs) to be more factual. We use biography generation as the task of our pilot study for two main reasons: (1) Biography generation is a simplified setting where factuality is the sole focus of the alignment process. As we will discuss in Section 4, studying factual alignment on diverse human instructions is more complex, as the alignment process encompasses aspects beyond factuality, such as helpfulness and safety. (2) Evaluating the factuality of biography generation is relatively easy since Wikipedia covers sufficient information for public figures and most of the facts about a person are non-debatable [Min et al., 2023].

### 3.1  Alignment for Biography Generation

A standard alignment procedure consists of supervised fine-tuning (SFT) and reinforcement learning (RL). In this pilot study, our main goal is to teach LLMs to generate biography with reduced misinformation. For the experiment, we compile training and evaluation datasets comprising 500 and 183 diverse human entities, respectively (further details provided in Appendix A.1). We employ FACTSCORE [FS; Min et al., 2023] as the automated metric for assessing factuality, given its fine-grained evaluation capabilities for long-form text generation and its strong correlation with human judgments.[3] To study factuality alignment in this pilot study, we posit that training data is needed where the responses are more factual than the LLM's own generations. Thus, we use retrieval-augmented LLMs [RAG; Lewis et al., 2020] to generate training data, which has been shown to output more factual responses [Mialon et al., 2023].

Throughout the paper, we refer to the pre-trained (PT), supervised fine-tuned (SFT), and direct preference optimization (DPO) fine-tuned LLMs as PT, SFT, and DPO, respectively.[4]

---

[3]We use the evaluator: `retrieval+llama+npm`

[4]Note that in our experiments, we use DPO as the substitute of RL [Schulman et al., 2017].

| Fact-based ($x \in X^{fact}$) | | Non fact-based ($x \notin X^{fact}$) | |
|---|---|---|---|
| (1) | Do you have any information about the Commodore 64? | (6) | How would a child feel if it fell down on the ground hitting its face? |
| (2) | Hi, could you help me to solve this cubic equation using Cardano's Mehod (step by step if possible), please? -> " x^3 + 2x^2 - x - 1 = 0 " | (7) | Write a fun story that can be told in 3 minutes at the dinner table. We are 3 developers enjoying a pizza. The story must contain these word: zombie, ethernet cable, sudo, dashboard. |
| (3) | Please give me a brief history of coffee. | (8) | Tell me a story about a pig who goes to the moon. |
| (4) | What are the principles at play in UHPLC-MS analysis? | (9) | Is the internet's focus on engagement the root of most of its problems and shortcomings? |
| (5) | Explain the significance of the American Revolution, including the events that led up to it, the impact it had on the world, and its ongoing relevance today. | (10) | Can you tell me a bit about what has gone into your creation? |

Figure 2: Instructions from Open Assistant dataset. The instructions are classified with SFT model using the prompt in Appendix Figure 4.

**SFT.** We explore two sources of supervision to generate training data (detailed in Appendix A.1): (1) using $PT^{RAG}$ with few-shot demonstration to generate biographies for each name entity in training data, where $PT^{RAG}$ is PT augmented with an off-the-shelf retriever [Lin et al., 2023]; (2) using vanilla PT with few-shot demonstration to generate training data as a baseline. As shown in Table 1, $PT^{RAG}$ is indeed much more factual than PT. However, a surprising discovery in the pilot study is that *fine-tuning on such more factual instruction–biography pairs generated by $PT^{RAG}$ results in a less factual* SFT *model* (row 4 vs 3).

**DPO.** We further fine-tune the LLMs to be more factual through DPO. An intuitive way to create factuality preference pairs is to directly use the samples from $PT^{RAG}$ and PT as positives and negatives since $PT^{RAG}$ generates more factual biographies than PT (row 2 vs 1). Another approach is to employ FACTSCORE (FS) as the reward to select positive and negative samples among the generations from PT itself [Tian et al., 2024] (detailed in Appendix A.1). As shown in Table 1, DPO fine-tuned on self-generated data with FS reward guides models to generate more factual responses (row 5 vs 3); however, DPO fine-tuned with the supervision of $PT^{RAG}$ makes the models hallucinate even more than its SFT counterpart (6 vs 4).

Table 1: Pilot study on bio generation. Pos. denotes the positives for SFT or DPO. Neg. denotes the negatives for DPO. FS denotes FACTSCORE.

| Llama-2 7B | src. of supervision | | Bio | |
|---|---|---|---|---|
| | Pos. | Neg. | FS | # Corr. / Err. |
| (1) PT | - | - | 39.1 | 14.4 / 22.0 |
| (2) $PT^{RAG}$ | - | - | 55.4 | 18.6 / 15.9 |
| (3) SFT | PT | - | 37.9 | 13.4 / 21.8 |
| (4) | $PT^{RAG}$ | - | 35.7 | 13.5 / 23.7 |
| (5) DPO | $PT^*$ | $PT^*$ | 41.6 | 15.4 / 20.7 |
| (6) | $PT^{RAG}$ | PT | 23.5 | 12.7 / 34.9 |

$^*$ FACTSCORE is used to select positives and negatives.

This outcome suggests that compelling models to generate responses akin to $PT^{RAG}$ prompts increases hallucination. Conversely, fine-tuning LLMs on their own generations appears to be crucial for factual alignment, a finding applicable to both SFT and DPO fine-tuning.

## 3.2 Strategies for Factual Alignment

From the pilot study, we find that better quality data (in terms of factuality) for SFT and DPO does not necessarily yield models with better factual alignment. This is likely because the supervision from RAG contains information unknown to the LLM; thus, fine-tuning on RAG generated responses may inadvertently encourage the LLM to output unfamiliar information. To avoid unknown knowledge from being presented to the LLM, a viable strategy is to create SFT and DPO training data using the generated responses from the LLM itself.

## 4 Factuality-Aware Alignment

In the section, we further extend our discussion of factual alignment to encompass more general instructions. Unlike biography generation in Section 3, where factuality is the main alignment objective, human instructions are diverse and complex, necessitating a range of alignment skill sets beyond factuality alone; e.g., logical thinking, problem handling and user alignment [Ye et al., 2024]. Thus, conducting factual alignment with the diverse instructions face two main challenges: (1) different instructions may demand distinct skill sets. For example, in Figure 2, instruction 3, "Please give me a brief history of coffee", necessitates factual accuracy and concise summarization,

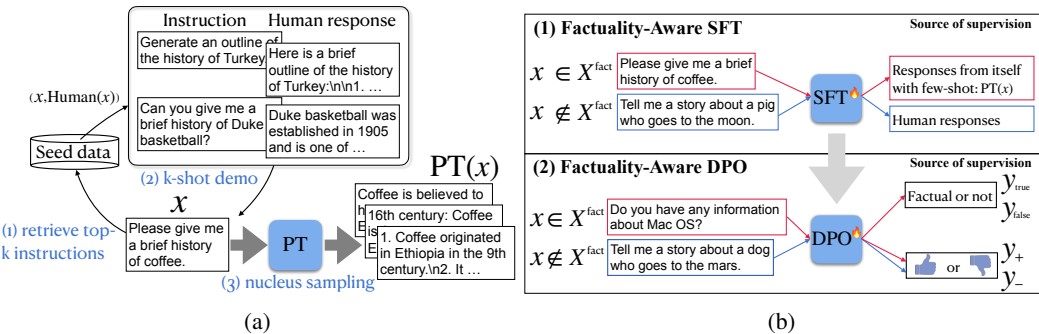

Figure 3: Illustrations of (a) response generation using a pre-trained LLM (PT) with few-shot demonstration; (b) factuality-aware alignment.

while instruction 8, "Tell me a story about a pig who goes to the moon", prioritizes creativity and imagination over strict factuality. (2) As recent studies have emphasized [Ye et al., 2024, Hosking et al., 2024], using a single scalar for reward modeling fails to adequately address multiple alignment skill sets and often under-presents the aspect of factuality.

To tackle the aforementioned challenges, we propose *factuality-aware alignment* (FLAME🔥). To address the first challenge, we propose to prompt LLMs to classify whether a given instruction demands the response to be factual, as shown in Figure 2. We then apply the factuality fine-tuning strategy for SFT and DPO discussed in Section 3.2 to those fact-based instructions. Furthermore, to address the second challenge, we employ separate rewards to evaluate the factuality and instruction-following capability of an LLM. For simplicity, our work only considers two alignment skill sets: instruction following and factuality. We leave more comprehensive reward modeling to future work.

In the following, we first describe our baseline alignment approach and introduce our proposed factuality-aware alignment built on top of the baseline alignment procedure.

## 4.1 Baseline Alignment

We initialize PT from Llama-2 70B pre-trained model[5] and build our baseline alignment procedure following self-rewarding language models [Yuan et al., 2024] due to its simplicity and independence of other strong LLMs (e.g., GPT4) or human evaluators as a reward model. The alignment comprises two steps: (1) building SFT model fine-tuned on a high-quality seed data consisting of 3,200 instructions and each instruction is paired with the best response created by humans from Open Assistant dataset [OASST; Köpf et al., 2023]; (2) further fine-tuning SFT through DPO on instruction-following preference data $(x, y_+, y_-)$ constructed by itself (SFT) as the reward model, $\text{RM}^{\text{IF}}$, where $y_+$ and $y_-$ are the positive and negative responses for a given prompt $x$, respectively. The resulting fine-tuned model is denoted as SFT + DPO. Note that, following Yuan et al. [2024], we use additional augmented 20K instructions to create the preference training data for DPO fine-tuning. Further details are provided in Appendix A.3.

## 4.2 Our Approach

### 4.2.1 Factuality-Aware SFT (SFT🔥)

Although leveraging human created high-quality seed data is a reasonable choice for SFT [Zhou et al., 2023], our study in Section 3 suggests that fine-tuning on such high-quality data generated by models other than the LLM itself may present unknown information to the LLM, which may in turn encourage hallucination. To address the above issue, for each instruction from the seed data, we elicit the knowledge from the pre-trained LM itself by generating the responses with a few-shot demonstration. Furthermore, to better use the knowledge from both humans and the pre-trained LLM itself, we propose to utilize human generated responses for non-fact-based instructions, while leveraging the responses sampled from pre-trained LLMs for fact-based instructions to mitigate the introduction of unknown knowledge.

---

[5] https://huggingface.co/meta-llama/Llama-2-70b

Specifically, we create factuality-aware alignment training data for SFT with two steps. (1) Classifying instructions: we first prompt SFT to judge whether an instruction from the seed data is fact-based ($x \in X^{\text{fact}}$) or not.[6] (2) Eliciting knowledge from PT: as illustrated in Figure 3(a), we sample 10 responses from PT with 5-shot demonstration, $(x_0, \text{Human}(x_0)) \cdots (x_4, \text{Human}(x_4))$, where $x_k$ is the top-$k$ similar instruction to $x$ retrieved by DRAGON+ [Lin et al., 2023] from the seed data. $\text{Human}(x_k)$ denotes the corresponding human response to $x_k$ in the seed data. As illustrated in Figure 3(b) (upper), the resulting training data for SFT is $(x \notin X^{\text{fact}}, \text{Human}(x)), (x \in X^{\text{fact}}, \text{PT}(x))$, where $\text{PT}(x)$ denotes the set of responses to $x$ sampled from PT. The fine-tuned model is denoted as $\text{SFT}^{\spadesuit}$.

### 4.2.2  Factuality-Aware DPO ($\text{DPO}^{\spadesuit}$)

At the second stage of alignment with DPO, we use $\text{SFT}^{\spadesuit}$ to generate multiple responses $y_0, y_1, \cdots$ for a given instruction $x$; then, using $\text{SFT}^{\spadesuit}$ itself as the reward model ($\text{RM}^{\text{IF}}$) to create a preference pair: $(x, y_+, y_-)$.[7] The above data creation procedure is the same as the second stage of our baseline alignment in Section 4.1. However, recent studies [Saha et al., 2024, Hosking et al., 2024, Ye et al., 2024] indicate that a single scalar reward from human feedback or LLM reward models may underrepresent the aspect of factuality. To address this limitation, we introduce another factuality reward model ($\text{RM}^{\text{fact}}$) to evaluate factuality of responses and create a factuality preference pair for fact-based instructions: $(x \in X^{\text{fact}}, y_{\text{true}}, y_{\text{false}})$.

Specifically, we build $\text{RM}^{\text{fact}}$ with retrieval augmentation to measure the percentage of facts in a response that are correct. $\text{RM}^{\text{fact}}$ comprises two main components: atomic fact decomposition and retrieval augmented claim verification. We detail the components and ablate their impacts on the quality of $\text{RM}^{\text{fact}}$ in Appendix A.5. We compute factuality reward for the same responses sampled from $\text{SFT}^{\spadesuit}$: $\text{RM}^{\text{fact}}(x, y_0), \text{RM}^{\text{fact}}(x, y_1), \cdots$. The response with the highest (lowest) factuality reward is chosen as $y_{\text{true}}$ ($y_{\text{false}}$). Note that if the chosen paired responses show large difference in instruction-following reward, we discard the pair; i.e., $|\text{RM}^{\text{IF}}(x, y_{\text{true}}) - \text{RM}^{\text{IF}}(x, y_{\text{false}})| > 0.5$. As illustrated in Figure 3(b) (lower), in factuality-aware DPO training, the model is initialized from $\text{SFT}^{\spadesuit}$ and the fine-tuned model is our final factuality-aware aligned model, denoted $\text{SFT}^{\spadesuit} + \text{DPO}^{\spadesuit}$. The specific procedures for fine-tuning models in both the SFT and DPO are described in Appendix A.6.

## 5  Experiments

### 5.1  Evaluation Datasets and Metrics

**Instruction Following.**    We use the 805 instruction-following tasks from Alpaca Eval [Dubois et al., 2024] to evaluate models head-to-head win rate against our baselines using the recommended evaluator: `alpaca_eval_gpt4_turbo_fn`. We use SFT and SFT + DPO described in Section 4.1 as the baselines for win rate comparisons.

**Factuality.**    We evaluate models on three datasets with diverse knowledge-intensive instructions for factuality. (1) Biography: a knowledge insensitive sub-task of instruction-following tasks. Following our pilot study in Section 3, we use the 183 human entities provided by Min et al. [2023] with the prompt "Tell me a bio of `entity name`". (2) Alpaca Fact: we extract the fact-based instructions from the 803 instructions using our SFT model (with the prompt shown in Appendix Figure 4), resulting in 241 instructions. (3) FAVA [Mishra et al., 2024][8]: the 141 knowledge-intensive instructions from multiple sources, including Open Assistant [Köpf et al., 2023], No Robots [Rajani et al., 2023], WebNLG [Gardent et al., 2017] and manually created datasets. We report FACTSCORE (FS) without length penalty as the metric for all the three datasets. Note that original FS computes proportion of correct facts with additional penalty on short generations with less than 10 atomic facts. This penalty aims to address situations where models provide insufficiently detailed answers. We assume that this aspect is considered in the evaluation of instruction following in Alpaca Eval. In addition, we also

---

[6]Prompt for fact-based instruction classification is shown in Appendix Figure 4.

[7]We sample 4 responses for each augmented instruction.

[8]https://huggingface.co/datasets/fava-uw/fava-data/blob/main/annotations.json

Table 2: Experimental results of supervised fine-tuning on Open Assistant dataset. PT denotes pre-trained Llama2 70B with 5-shot demonstration. SFT$^{\text{fact}}$ denotes the variant which only optimizes factuality. FS denotes FACTSCORE.

| Llama-2 70B | src. of supervision | | Alpaca Eval | Bio | | Alpaca Fact | | FAVA | |
|---|---|---|---|---|---|---|---|---|---|
| | Human | PT | win rate over (1) | FS | # Corr. / Err. | FS | # Corr. / Err. | FS | # Corr. / Err. |
| (0) PT | - | - | | 53.1 | 15.3 / 13.5 | - | - | - | - |
| (1) SFT | ✓ | ✗ | 50.0 | 44.7 | 21.1 / 26.8 | 38.6 | 16.7 / 29.0 | **54.4** | 21.2 / 25.8 |
| (2) SFT$^{\text{fact}}$ | ✗ | ✓ | 48.1 | 48.5 | 19.6 / 20.6 | **42.0** | 17.5 / 28.4 | 53.3 | 18.3 / 24.2 |
| (3) SFT$^{\clubsuit}$ | ✓* | ✓* | **51.2** | **49.5** | 19.9 / 19.5 | 41.4 | 18.3 / 27.7 | 54.2 | 19.3 / 22.4 |

* SFT$^{\clubsuit}$ uses supervision from Human and PT for non-fact-based and fact-based instructions, respectively.

Table 3: Experiments of direct preference optimization (DPO). IF. and Fact. denote instruction following $(x, y_+, y_-)$ and factuality $(x \in X^{\text{fact}}, y_{\text{true}}, y_{\text{false}})$ preference data, where $X^{\text{fact}}$ denotes the set of fact-based instructions. DPO$^{\text{fact}}$ denotes the variant which only optimizes factuality. The preference data statistics is listed in Appendix, Table 11.

| Llama-2 70B | src. of supervision | | Alpaca Eval | Bio | | Alpaca Fact | | FAVA | |
|---|---|---|---|---|---|---|---|---|---|
| | IF. | Fact. | win rate over (2) | FS | # Corr. / Err. | FS | # Corr. / Err. | FS | # Corr. / Err. |
| (0) Chat | Proprietary data | | 66.2 | 33.2 | 23.4 / 43.6 | 39.3 | 22.3 / 36.4 | 47.5 | 28.0 / 31.3 |
| (1) SFT | - | - | 27.1 | 44.7 | 21.1 / 26.8 | 38.6 | 16.7 / 29.0 | 54.4 | 21.2 / 25.8 |
| (2) + DPO | ✓ | ✗ | 50.0 | 42.3 | 24.6 / 35.0 | 41.6 | 22.9 / 34.6 | 52.9 | 28.1 / 26.8 |
| (3) + DPO$^{\text{fact}}$ | ✗ | ✓ | 40.8 | 47.1 | 19.8 / 23.9 | 48.2 | 17.5 / 19.0 | 57.9 | 20.0 / 15.9 |
| (4) + DPO$^{\clubsuit}$ | ✓ | ✓ | **51.7** | 44.9 | 23.7 / 30.3 | 45.0 | 23.1 / 28.7 | 56.4 | 27.1 / 23.3 |
| (5) SFT$^{\clubsuit}$ | - | - | 29.1 | **49.5** | 19.9 / 19.5 | 41.4 | 18.3 / 27.7 | 54.2 | 19.3 / 22.4 |
| (6) + DPO | ✓ | ✗ | 50.4 | 46.3 | 24.0 / 28.7 | 43.9 | 21.6 / 28.8 | 55.0 | 25.4 / 22.0 |
| (7) + DPO$^{\clubsuit}$ | ✓ | ✓ | 51.2 | 47.9 | 25.9 / 28.5 | **48.7** | 24.1 / 25.5 | **58.9** | 29.0 / 22.2 |

report the number of correct and erroneous facts. All the numbers reported are averaged over the instructions in each dataset.

In addition, we also evaluate our fine-tuned models' truthfulness using TruthfulQA [Lin et al., 2022]. We evaluate model performance in the generation task and use ROUGE [Lin, 2004] and BLEU [Papineni et al., 2002] to measure the quality of responses.

## 5.2 Comparisons of SFT

Table 2 compares the pre-trained Llama-2 70B fine-tuned on OASST dataset with responses from different sources. We list the FACTSCORE (FS) of biography generation using the pre-trained model through Bio 5-shot demonstration as reference (row 0) and SFT, which is fine-tuned on our seed data with human-created responses, is our baseline (row 1). We first notice that SFT shows significant FACTSCORE degradation (53.1 vs 44.7) compared to Bio 5-shot with the pre-trained model. It seems that SFT tends to generate more lengthy responses but with more erroneous facts.

When eliciting the knowledge from PT by fine-tuning on its own generated responses, SFT$^{\text{fact}}$ generates more factual responses in Biography and Alpaca (row 2 vs 1). However, it shows slightly inferior instruction-following capability in Alpaca Eval. This result demonstrates that human responses indeed teach LLMs how to better follow instructions but also encourage LLMs to output more false facts. On the other hand, eliciting the knowledge from the pre-trained model itself avoids the encouragement of hallucination albeit with a slight reduction in instruction-following capability. Finally, SFT$^{\clubsuit}$ combining supervision from humans and PT, shows comparable instruction-following capability and output more factual responses on fact-based instructions (row 3 vs 1).

## 5.3 Comparisons of DPO

Table 3 compares different DPO training recipes. First, we conduct DPO fine-tuning on our SFT baseline, SFT. When further aligning the model to follow instructions, DPO sees a significant improvement in instruction-following capability (row 2 vs 1) with win rate 72.9 over SFT; however, the instruction aligned model tends to output lengthy responses with more factual errors (see examples in Appendix Figure 10). On the other hand, when only aligned with factual preference data, DPO$^{\text{fact}}$ shows less improvement in instruction-following capability (row 1 vs 3). These results indicate that

preference optimization for either instruction following or factuality alone may come at the expense of the other since the former encourages models to output long and detailed responses while the later discourages models to output false claims. When jointly conducting instruction and factuality alignment, DPO♦ not only better follows instructions but also outputs more factual responses (row 4 vs 1, 2). Finally, initializing from SFT♦, the DPO fine-tuned models are more factual than their counterparts (i.e., 6 vs 2 and 7 vs 4) without instruction-following capability degrade. We also list the results from Llama-2-Chat 70B (row 0) and observe that despite of its strong instruction-following capability, it tends to output many more incorrect facts. These results demonstrate that standard alignment, even on proprietary commercial data, may encourage LLMs to hallucinate. In contrast, our factuality-aware alignment guides LLMs to output more factual responses without degradation in their general instruction-following capabilities.

It is worth noting that SFT$^{fact}$ and DPO$^{fact}$ are similar to SFT and DPO fine-tuning proposed by Tian et al. [2024], which improve LLMs' factuality but degrade their instruction-following capability. Also, we do not observe our SFT and DPO variants outperform the pre-trained model with few-shot demonstrations on biography generation (row 0 in Table 2. This is possibly due to the alignment tax found in previous work [Ouyang et al., 2022], which degrades LLMs' accuracy on the standard knowledge benchmarks. How to improve both models' instruction-following capability and their accuracy on standard knowledge benchmarks is worth exploring, which we leave for future work.

## 5.4 Results on TruthfulQA

Table 4 compares models performance on TruthfulQA. Generally, we observe that our factuality-aware alignment training guides LLMs to output more truthful responses. For example, factuality-aware SFT improves LLMs' truthfulness (row 5 vs 1). In addition, DPO fine-tuning on the factuality preference data guides LLMs to output more truthful responses (rows 3,4 vs 2 and 7 vs 6). Note that we observe that SFT and DPO models show a reverse trend in BLUE and ROUGE. This is likely because SFT models tend to generate shorter responses than the DPO ones do.

In addition, Table 5 reports models' accuracy in tasks of multiple choices from TruthfulQA. No significant differences between models are observed. This is possibly because we mainly focus on the tasks of long-form response generation while TruthfulQA-MC task is formed by short-form answers. The discrepancy between improving LLMs' factuality on long-form and short-form generation is also found by the previous work [Chuang et al., 2024]. Appendix Table 9 reports more evaluation results on other NLP benchmarks.

Table 4: Results on TruthfulQA.

| Llama-2 70B | src. of supervision | | TruthfulQA | |
| --- | --- | --- | --- | --- |
| | IF. | Fact. | BLUE | ROUGE |
| (0) Chat | Proprietary data | | 0.21 | 1.16 |
| (1) SFT | - | - | 0.37 | 0.20 |
| (2) + DPO | ✓ | ✗ | 0.03 | 0.54 |
| (3) + DPO$^{fact}$ | ✗ | ✓ | 0.30 | 1.12 |
| (4) + DPO♦ | ✓ | ✓ | 0.15 | 0.80 |
| (5) SFT♦ | - | - | 0.39 | 0.51 |
| (6) + DPO | ✓ | ✗ | 0.07 | 0.91 |
| (7) + DPO♦ | ✓ | ✓ | 0.20 | 0.96 |

Table 5: Results on TruthfulQA multiple choices.

| Llama-2 70B | src. of supervision | | TruthfulQA-MC | | |
| --- | --- | --- | --- | --- | --- |
| | IF. | Fact. | MC1 | MC2 | MC3 |
| (0) Chat | Proprietary data | | 32.2 | 50.2 | 25.4 |
| (1) SFT | - | - | 30.8 | 45.7 | 23.9 |
| (2) + DPO | ✓ | ✗ | 30.5 | 46.0 | 23.4 |
| (3) + DPO$^{fact}$ | ✗ | ✓ | 31.8 | 46.8 | 24.3 |
| (4) + DPO♦ | ✓ | ✓ | 30.8 | 46.0 | 23.6 |
| (5) SFT♦ | - | - | 29.9 | 44.8 | 22.5 |
| (6) + DPO | ✓ | ✗ | 31.5 | 47.0 | 24.0 |
| (7) + DPO♦ | ✓ | ✓ | 30.5 | 45.4 | 23.1 |

## 6 Discussion

### 6.1 Effects of Fact-Based Instruction Classification

In our factuality-aware alignment, we prompt SFT to judge whether an instruction requires a factual response and apply our factuality alignment strategy to the fact-based instruction. Without the instruction classification, in our factuality-aware SFT, we cannot create supervision from Human and PT responses for respective non-fact-based and fact-based instructions. Instead, for each instruction, we create instruction–response pairs from 1 and 10 responses from Human and PT as supervisions, respectively. Note that, during fine-tuning, for each instruction, we randomly sample instruction–response pair either created from Human or PT with same probability. The SFT model shows

degradation in both instruction-following capability and factuality results, as shown in row 1 vs 2 of Table 6. Second, for factuality-aware DPO, without the instruction classification, we create factuality preference pairs from all instructions instead of fact-based instructions. The DPO fine-tuned model outputs slightly more factual responses but sacrifice instruction-following capability, as shown in row 3 vs 4 of Table 6.

## 6.2 Effects of Fact-Based Sentence Classification

In addition, we observe that not all the sentences in a response to a fact-based instruction require fact check. For example, given the response, "Of course. The Commodore 64 is a 8-bit home computer that was released by Commodore International in August 1982.", conducting fact check for the first sentence "Of course." is not necessary and may make the factuality reward less accurate. To address this issue, we prompt SFT to judge whether each sentence in a response required fact check using the prompt in Appendix Figure 6. We only conduct fact check and compute factu-

Table 6: Effects of fact-based classification.

| | | Classifier | | **Alpaca Eval** | **Bio** | |
|---|---|---|---|---|---|---|
| | | Inst. | Sent. | win rate | FS | # Corr. / Err. |
| (1) | SFT♦ | ✗ | - | 47.6* | 48.4 | 20.5 / 21.4 |
| (2) | | ✓ | - | 51.2* | 49.5 | 19.9 / 19.5 |
| (3) | | ✗ | ✗ | 46.8△ | 46.8 | 21.7 / 25.3 |
| (4) | SFT + DPO♦ | ✓ | ✗ | 51.7△ | 45.0 | 23.7 / 30.3 |
| (5) | | ✓ | ✓ | 51.3△ | 42.9 | 25.5 / 36.8 |

* comparing with SFT baseline, SFT.
△ comparing with DPO baseline, SFT + DPO.

ality rewards for those fact-based sentences. However, as shown in Table 6, computing factuality rewards for fact-based sentences makes our factual alignment less effective (row 5 vs 4). This is likely because the fact-based sentence classifier is not accurate enough and brings noise into our factuality reward model (see examples in Appendix Figure 7).

## 6.3 Ablations on Factuality Preference Data Creation

In this section, we examine different ways of creating factuality preference data for factuality-aware DPO training. First, for each fact-based instruction, instead of choosing the responses (among the 4 generated responses) with the maximum and minimum factuality rewards ($\mathrm{RM}^{\mathrm{fact}}$) as the respective positive and negative samples, we enumerate all the possible response pairs and choose the response with higher (lower) $\mathrm{RM}^{\mathrm{fact}}$ as the positive (negative) sample from each enu-

Table 7: Ablation on factuality preference data.

| Factuality preference data | | | **Alpaca Eval** | **Bio** |
|---|---|---|---|---|
| Reward model | Pos.,Neg. | # pairs | win rate△ | FS |
| $\mathrm{RM}^{\mathrm{fact}}$ | max, min | 3,315 | 51.7 | 44.9 |
| $\mathrm{RM}^{\mathrm{fact}}$ | enum. | 5,126 | 50.7 | 45.0 |
| $\mathrm{RM}^{\mathrm{IF}} + 5*\mathrm{RM}^{\mathrm{fact}}$ | max, min | 6,340 | 50.1 | 45.1 |

△ comparing with DPO baseline, SFT + DPO.

merated pair. If the difference of $\mathrm{RM}^{\mathrm{fact}}$ is smaller than $0.2$, we treat them as equal and discard the pairs. Note that for both row 1 and 2 in Table 7, we also discard the pairs with the difference of instruction-following rewards ($\mathrm{RM}^{\mathrm{IF}}$) larger than $0.5$ (as mentioned in Section 4.2.2). Alternatively, for each response, we linearly combine the rewards, $\mathrm{RM}^{\mathrm{IF}}$ (1–5 scale) and $\mathrm{RM}^{\mathrm{fact}}$ (0–1 scale), with the respective weight of 1 and 5 as a composite reward. For each instruction, we choose the responses with the maximum and minimum composite rewards as the positive and negative. As a result, both data creation approaches increase the number of factuality preference pairs; however, they yield trivial improvement in factuality but slight degrade in instruction following (rows 2, 3 vs 1). This result also indicates that leveraging a single reward model, which can be incorporated with PPO [Schulman et al., 2017], is possible to improve both models' instruction-following capability and factuality

## 6.4 Impacts of DPO on Generation Length

Table 8 lists the average length of models' responses for each dataset. We observe that DPO fine-tuned models tend to output lengthy responses than SFT except for DPO$^{\mathrm{fact}}$ on Biography. This trend indicates that our instruction-following reward model $\mathrm{RM}^{\mathrm{IF}}$ guides LLMs to output more detailed and lengthy responses. In addition, we observe that although DPO♦ out-

Table 8: Effects of DPO on response length.

| | **Alpaca Eval** | **Bio** | **Alpaca Fact** | **FAVA** |
|---|---|---|---|---|
| (1) SFT | 897 | 1221 | 969 | 912 |
| (2) + DPO | 1470 | 1494 | 1586 | 1540 |
| (3) + DPO$^{\mathrm{fact}}$ | 1160 | 1166 | 1192 | 1104 |
| (4) + DPO♦ | 1474 | 1395 | 1528 | 1422 |

puts responses with similar length as DPO on Alpaca Eval, DPO generates a slightly shorter responses for the fact-based instructions in the other three datasets. This results show that our factuality-aware DPO training mainly impacts models' responses for fact-based instructions. The impact is mainly to reduce the false claims, evidenced by the numbers of erroneous facts in rows 2 and 4 of Table 3).

### 6.5 Case Studies

Figure 10 (in Appendix) showcases the generations of different models, SFT, SFT + DPO and SFT + DPO, on Alpaca Eval and Biography. Given the instruction, "What are the names of some famous actors that started their careers on Broadway?", SFT only lists some names of Broadway actors while DPO fine-tuned models generate detailed information for each listed Broadway actor. As for biography generations, we observe that given the instruction to generate a biography for a rare name entity, Marianne McAndrew, SFT + DPO generates a detailed response but with many wrong facts while SFT and SFT + DPO give relatively short responses. For the frequent entity, Ji Sung, all the models generate detailed and mostly correct responses. This qualitative analysis shows that SFT + DPO tends to generate detailed responses for most instructions, but for those instructions required tailed knowledge (e.g., rare entity) likely unknown to LLMs [Mallen et al., 2023], it reduces erroneous facts by giving less detailed responses, which is also observed by Kang et al. [2024].

## 7 Conclusion and Future Work

In this paper, we present a study to enhance the factuality of large language models (LLMs). We first identify that the standard alignment approach, comprising SFT and RLAIF with DPO, may inadvertently encourage LLMs to produce more erroneous facts. Specifically, during the SFT stage, fine-tuning LLMs with high-quality human responses may introduce unfamiliar information, prompting LLMs to output unknown facts. Additionally, during the DPO stage, enhancing LLMs' ability to follow instructions may result in more detailed and lengthy responses but often leads to increased hallucination. To tackle the shortcomings of the standard alignment, we propose a factuality-aware alignment method, which includes factuality-aware SFT and DPO. Quantitative and qualitative analyses demonstrate that our factuality-aware alignment not only guides LLMs to generate detailed and helpful responses but also helps prevent the generation of false claims.

While we have successfully integrated factuality into standard alignment procedure, our work only considers two alignment skill sets: instruction following (or helpfulness) and factuality. In practice, each instruction may require consideration of multiple and distinct alignment skill sets [Saha et al., 2024]. The method to optimize for these skill sets tailored to each query requires further study. In our experiments, we note that optimizing preferences solely for instruction following or factuality could potentially compromise the other. While our factuality-aware alignment demonstrated improvements in both aspects, it is uncertain whether there is a trade-off between the two aspects when integrating our approach to large-scale alignment [Touvron et al., 2023]. Finally, as shown in Appendix Figure 7, not all the claims (or sentences) in a response require fact verification, a more accurate factuality reward model should take this factor into account. While our preliminary experiment, which removes non-fact-based sentences from the factuality reward modeling (Section 6.2), shows suboptimal performance, we believe that further study can bring more insights.

## Acknowledgements

We thank Bhargavi Paranjape for sharing fine-tuned Llama-2 7B for atomic fact decomposition and Jing Xu, Weizhe Yuan and Jason Weston for their helpful suggestions.

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

# A Appendix

## A.1 Biography Data Generation

**Entities for Training and Evaluation.** We use 500 diverse human entities to create training data for SFT and DPO; then, evaluate LLMs' generation factuality on another 183 human entities from Min et al. [2023].[9] Note that the human entities for training and evaluation are uniformly sampled from entities across diverse nationalities, professions, and rarities. The instruction is generated with the format: Tell me a bio of `entity name`.

**Creating Training Data for SFT.** We randomly sample 5 human entities among the 500 entities for training and generate their biographies using Llama-2-Chat 70B as 5-shot demonstration.[10] With the 5-shot demonstration, we use pre-trained Llama-2 7B to generate 10 biographies for each human entity from the remaining 495 ones.[11] We set temperature 0.7 and top-p 0.9 when generate multiple responses from LLMs in all our experiments. We use the created 4,950 name entity–biography pairs to fine-tune the pre-trained Llama-2 7B. As for generating training data with RAG, we prepend the top-10 passages from our retrieval system (detailed in Appendix A.2) to each instruction and generate 10 biographies for each entity from RAG with 5-shot demonstrations. Note that we only prepend top-1 passage for each instruction in the demonstration.

**Creating Factuality Preference Pairs for DPO.** To construct factuality preference pairs, we first compute FACTSCORE (FS) for all the 4,950 biographies previously created by PT. Then, for each name entity, we compare the FS for all the possible 45 pairs from the 10 generated biographies and construct DPO pairs using the biography with a higher (lower) FS as a positive (negative). Note that we discard the pairs if they show tied FS.

## A.2 Retrieval Models

For each query, we retrieve top-20 candidate passages from Wikipedia using DRAGON+ [Lin et al., 2023] and re-rank the candidates using a 12-layer cross-encoder[12]. We use the Wikipedia version from the Dec. 20, 2021 dump released by Izacard et al. [2023] in this work.

## A.3 Alignment with Self Rewarding

**SFT.** At SFT stage, we fine-tune PT on two seed datasets: (1) Instruction-following training (IFT) data from Li et al. [2024], consisting of 3200 instruction–response pairs created by humans from Open Assistant dataset [OASST; Köpf et al., 2023], where we only use the first conversational turns in the English that are annotated rank 0;[13] (2) evaluation following training (EFT) data from Yuan et al. [2024], the LLM-as-a-Judge data consists of 1630 samples, each of which contains instruction, human response and the corresponding score of 1-5 scale (with chain-of-though evaluation reasoning): $(x, y, r)$, where $(x, y)$ pairs are also selected from OASST other than training pairs and $r$ is created by the model fine-tuned only on IFT with manual filtering. The purpose of EFT is to enhance a LLM's capability as a reward model to judge the quality of a response in terms of relevance, coverage, usefulness, clarity and expertise. We refer readers to Yuan et al. [2024] for how EFT is created and filtered with minimum human efforts. The prompt template for LLM-as-a-Judge in EFT and an EFT training sample are shown in Appendix, Figure 8 and 9. We refer the baseline model fine-tuned on the IFT and EFT datasets as SFT.

**DPO for Instruction Following.** At the subsequent preference learning with DPO, following Wang et al. [2023a], we augment additional 20K instructions with Llama-2 70B chat model.[14] For each augmented instruction $x$, we use SFT to generate 4 responses and evaluate how well the responses follow the instruction with score of 1–5 scale: $\text{RM}^{\text{IF}}(x, y_0) \cdots ; \text{RM}^{\text{IF}}(x, y_3)$, where

---

[9] https://github.com/shmsw25/FActScore
[10] https://huggingface.co/meta-llama/Llama-2-70b-chat-hf
[11] https://huggingface.co/meta-llama/Llama-2-7b
[12] https://huggingface.co/sentence-transformers/all-MiniLM-L12-v2
[13] https://huggingface.co/datasets/OpenAssistant/oasst1
[14] https://huggingface.co/meta-llama/Llama-2-70b-chat-hf

$y_0, \cdots, y_3 \in \mathrm{SFT}(x)$ and $\mathrm{RM}^{\mathrm{IF}}$ is the instruction-following reward model. Note that, in self-rewarding [Yuan et al., 2024], $\mathrm{RM}^{\mathrm{IF}}$ is the same as SFT model. In addition, for each instruction–response pair, we use the same prompt in EFT seed data to sample the chain-of-thought evaluation three times and average the scores as the reward. Finally, for each instruction, we use the response with the highest (lowest) reward as the positive (negative) sample to form a preference pair for DPO training: $(x, y_+, y_-)$. We discard the pair, if $\mathrm{RM}^{\mathrm{IF}}(x, y_+) = \mathrm{RM}^{\mathrm{IF}}(x, y_-)$. In the DPO training, the model is initialized from SFT and the fine-tuned model is denoted $\mathrm{SFT} + \mathrm{DPO}$.

### A.4 More Evaluation Results on Standard Benchmarks

Table 9 compares the instruction fine-tuned models' (w/o and w/ involving FLAME) accuracy in tasks of MMLU [Hendrycks et al., 2021] and GSM8K [Cobbe et al., 2021]. A slight drop from FLAME is observed. This is possibly because we mainly focus on the tasks of long-form response generation while MMLU and GSM8K are the

Table 9: Results on MMLU and GSM8K.

| Llama-2 70B | src. of supervision | | **MMLU** | **GSM8K** |
| | IF. | Fact. | | |
|---|---|---|---|---|
| (1) SFT+ DPO | ✓ | ✗ | 69.3 | 59.3 |
| (2) SFT♦+ DPO♦ | ✓ | ✓ | 69.1 | 58.2 |

benchmarks with short-form answers. The discrepancy between improving LLMs' factuality on long-form and short-form generation is also found by the previous work [Chuang et al., 2024].

### A.5 Factuality Reward Modeling

**Factuality Reward Models.** We build a reward model $\mathrm{RM}^{\mathrm{fact}}$ to measure the factuality of each response. The factuality reward model consists of two main modules. (1) fact decomposition: we first use `nltk.tokenize` to split a response into sentences; then, use our Llama-2 7B model fine-tuned on public datasets [Liu et al., 2023, Chen et al., 2022, Malaviya et al., 2023] to conduct atomic fact decomposition for each sentence.[15] (2) Retrieval augmented claim verification: for each decomposed fact (or claim), we use the instruct Llama 7B fine-tuned on Super Natural Instructions [Wang et al., 2022] to do fact check with the prompt shown in Figure 5.[16] We append 10 retrieved supports (using the instruction as query) from our retrieval and re-ranking pipeline in Appendix A.2. Then, we compute the proportion of correct atomic facts in a response as a factuality reward.

**Quality of Factuality Reward Models.** We conduct ablation study on our factuality reward models. Specifically, we use our factuality reward models to detect the number of error facts in each instruction–response pair. We try different models for fact check using the prompt shown in Figure 5 with different numbers of retrieved supports. We use the LLMs' generated responses with human annotated hallucination provided by Mishra et al. [2024] to evaluate the quality of the factuality reward models.[17] Specifically, we rank the responses by numbers of errors detected and calculate the Kendall rank correlation ($\tau$)

Table 10: A comparison of factuality reward models. $\tau$ denotes the correlation between human annotation.

| | fact check model | # sup. | fact unit | $\tau$ |
|---|---|---|---|---|
| (1) | Instruct Llama 7B | 5 | atom. | 0.32 |
| (2) | | 10 | | 0.34 |
| (3) | SFT (Llama-2 70B) | 5 | atom. | 0.28 |
| (4) | | 10 | | 0.31 |
| (5) | Instruct Llama 7B | 5 | sent. | 0.20 |
| (6) | | 10 | | 0.25 |

between the rank lists by our factuality reward models and humans. As shown in Table 10, conducing fact check with more retrieved supports improves the accuracy of the factuality reward models (row 2 vs 1). In addition, our SFT, only fine-tuned on the IFT and EFT data, is capable of doing fact check, compared to Instruct Llama 7B fine-tuned on Super Natural Instructions [Wang et al., 2022]. Finally, instead of computing the number of error facts from decomposed atomic facts, we conduct fact check directly for each sentence in a response and calculate the number of false sentences as error facts. However, the quality of the reward models shows significant decrease (rows 5,6 vs 1,2). We finally adopt row 2 as our factuality reward model.

---

[15]With few-shot demonstration, SFT is able to decompose a sentence into atomic facts with acceptable accuracy. Fine-tuning a Llama-2 7B is to reduce the inference time.

[16]https://huggingface.co/kalpeshk2011/instruct-llama-7b-wdiff

[17]https://huggingface.co/datasets/fava-uw/fava-data/blob/main/annotations.json

Table 11: Training data statistics for different variants. IF. and Fact. denote instruction following $(x, y_+, y_-)$ and factuality $(x \in X^{\text{fact}}, y_{\text{true}}, y_{\text{false}})$ preference data, where $X^{\text{fact}}$ denotes the set of fact-based instructions.

| model variant | Seed IFT (# of Inst.) | | Preference (# of pairs) | |
|---|---|---|---|---|
| | $x \notin X^{\text{fact}}$ | $x \in X^{\text{fact}}$ | IF. | Fact. |
| SFT + DPO | | | 18,454 | - |
| SFT + DPO$^{\text{fact}}$ | 2,187 | 1,013 | - | 3,315 |
| SFT + DPO🔥 | | | 18,454 | 3,315 |
| SFT🔥 + DPO | 2,187 | 1,013 | 18,603 | - |
| SFT🔥 + DPO🔥 | | | 18,603 | 4,211 |

## A.6 Training Details

We fine-tune our models for 500 steps with a batch size of 32 and 64 on respective SFT and DPO stages. The learning rate and maximum sequence length is set to $1e - 6$ (which decays to $1e - 7$) and 2048, respectively. At SFT stage, we mix the IFT and EFT while at DPO stage, we set $\beta = 0.1$ and uniformly sample between self rewarding $(x, y_+, y_-)$ and factuality reward $(x, y_{\text{true}}, y_{\text{false}})$ preference data. Note that SFT (SFT🔥) + DPO meaning that we use SFT (SFT🔥) to create preference data, serve as instruction-following reward model $\text{RM}^{\text{IF}}$ and as the initialization of DPO. The data used to fine-tune different variants are listed in Table 11. We conduct fine-tuning with full parameters on 64 NVIDIA A100 (80GB) GPUs. SFT and DPO require around 1 and 4 hours to complete, respectively.

## A.7 Limitations

**Limitations.** In this paper, we use FACTSCORE as the main metric of factuality. However, as highlighted by Min et al. [2023], "FACTSCORE focuses on factual precision is only one aspect of the broader factuality problem." In addition, the validity of applying FACTSCORE to broader domains other than Biography has not been carefully studied. In our experiments, we can only claim that models' generation shows high agreement with the facts from Wikipedia, without considering whether the claims are open-ended and debatable. Finally, reward hacking is the potential issue of LLM alignment [Chen et al., 2024b] but the risk of reward hacking is still unknown when considering factuality.

## A.8 Broader Impacts

Increased factual accuracy in AI-generated responses may contribute to building trust in AI technology. Users are more likely to trust and rely on AI systems if they consistently provide accurate information, leading to broader adoption and acceptance of AI-driven solutions. However, FLAME does not deal with the facts which are open-ended and debatable and uses Wikipedia as the only source of facts. This may inadvertently reinforce some biases in the AI system's responses, potentially perpetuating societal inequalities.

```
[INST] Here is a question from a user: "<QUESTION>". To answer the above question, do you need the factual knowledge from
Wikipedia? Give an answer using the format: "Answer: Yes or No". [/INST]
```

Figure 4: Prompt to check whether an instruction is fact-based.

```
[INST] Answer the question about '<Instruction>' based on the given context.\n\n'<Retrieved Supports>'\n\nInput:
'<ATOMIC_FACT or SENTENCE>' True or False?\nOutput: [/INST]
```

Figure 5: Prompt for fact check.

```
[INST] Does the given sentence contain factual knowledge from Wikipedia? Question: '<QUERY>'; Sentence: '<CLAIM>'. Answer
with Yes or No. [/INST]
```

Figure 6: Prompt to check whether a claim is fact-based.

Figure 7: The results of whether a sentence is fact-based or not classified by SFT with prompt in Figure 6.

```
[INST] Review the user's question and the corresponding response using the additive 5-point scoring system described below. Points
are accumulated based on the satisfaction of each criterion:
- Add 1 point if the response is relevant and provides some information related to the user's inquiry, even if it is incomplete or
contains some irrelevant content.
- Add another point if the response addresses a substantial portion of the user's question, but does not completely resolve the
query or provide a direct answer.
- Award a third point if the response answers the basic elements of the user's question in a useful way, regardless of whether it
seems to have been written by an AI Assistant or if it has elements typically found in blogs or search results.
- Grant a fourth point if the response is clearly written from an AI Assistant's perspective, addressing the user's question
directly and comprehensively, and is well-organized and helpful, even if there is slight room for improvement in clarity,
conciseness or focus.
- Bestow a fifth point for a response that is impeccably tailored to the user's question by an AI Assistant, without extraneous
information, reflecting expert knowledge, and demonstrating a high-quality, engaging, and insightful answer.

User: <INSTRUCTION_HERE>

<response><RESPONSE_HERE></response>

After examining the user's instruction and the response:
- Briefly justify your total score, up to 100 words.
- Conclude with the score using the format: "Score: <total points>"
Remember to assess from the AI Assistant perspective, utilizing web search knowledge as necessary. To evaluate the response in
alignment with this additive scoring model, we'll systematically attribute points based on the outlined criteria. [/INST]
```

Figure 8: Prompt to evaluate models' instruction-following capability from Yuan et al. [2024].

[INST] Review the user's question and the corresponding response using the additive 5-point scoring system described below. Points are accumulated based on the satisfaction of each criterion:\n\n- Add 1 point if the response is relevant and provides some information related to the user's inquiry, even if it is incomplete or contains some irrelevant content.\n- Add another point if the response addresses a substantial portion of the user's question, but does not completely resolve the query or provide a direct answer.\n- Award a third point if the response answers the basic elements of the user's question in a useful way, regardless of whether it seems to have been written by an AI Assistant or if it has elements typically found in blogs or search results.\n- Grant a fourth point if the response is clearly written from an AI Assistant's perspective, addressing the user's question directly and comprehensively, and is well-organized and helpful, even if there is slight room for improvement in clarity, conciseness or focus.\n- Bestow a fifth point for a response that is impeccably tailored to the user's question by an AI Assistant, without extraneous information, reflecting expert knowledge, and demonstrating a high-quality, engaging, and insightful answer.\n\nUser: Write a series of steps explaining how Edward Snowden leaked classified government documents to the press, starting from when he first got employed by the U.S. government and ending shortly after he landed in Russia. \n\n<response>Here is a series of steps outlining the events surrounding Edward Snowden's leaking of classified government documents:\n\n Edward Snowden was employed by the U.S. government as a contractor for the National Security Agency (NSA) in 2006.\n\n Over the next several years, Snowden worked on various projects for the NSA and gained access to highly classified information and documents.\n\n In early 2013, Snowden became increasingly concerned about the government's widespread surveillance activities and felt that the public had a right to know about them.\n\n Snowden began copying and collecting classified documents from the NSA's systems, taking care to cover his tracks.\n\n In May 2013, Snowden left his job in Hawaii and traveled to Hong Kong, where he met with journalists from The Guardian newspaper and provided them with a selection of the classified documents he had obtained.\n\n The Guardian published a series of articles based on the leaked documents, starting in June 2013, which revealed the extent of the NSA's surveillance activities, including its collection of metadata on millions of American citizens and its access to the servers of major tech companies like Google and Facebook.\n\n The U.S. government charged Snowden with violating the Espionage Act and sought his extradition from Hong Kong.\n\n Fearing for his safety and freedom, Snowden left Hong Kong and traveled to Moscow, where he was granted asylum by the Russian government.\n\n Snowden has been living in Russia since 2013 and continues to speak out against government surveillance and advocate for privacy and civil liberties.\n\nThese are the major steps in the story of Edward Snowden's leaking of classified government documents to the press. The aftermath of these events continues to shape the ongoing debates around privacy, security, and government surveillance. </response>\n\nAfter examining the user's instruction and the response:\n\n- Briefly justify your total score, up to 100 words.\n- Conclude with the score using the format: \"Score: <total points>\"\n\nRemember to assess from the AI Assistant perspective, utilizing web search knowledge as necessary. [/INST] To evaluate the response in alignment with this additive scoring model, we'll systematically attribute points based on the outlined criteria.\n\n1. Relevance: 1 point - The response effectively addresses the user's question by outlining the key steps involved in Edward Snowden's disclosure of classified documents.\n2. Substantial portion: 1 point - The response comprehensively covers the essential events and elements of Snowden's story, from his employment to his asylum in Russia.\n3. Useful answer: 1 point - The answer delivers a well-structured account of Snowden's actions, government surveillance, and the consequences of his disclosures, providing valuable information for the user.\n4. AI Assistant perspective: 1 point - The response is written in a clear and organized manner, effectively summarizing the events surrounding Snowden's leaks.\n5. Expert knowledge and engaging answer: 1 point - The response demonstrates a good understanding of the topic, concisely presenting complex information and highlighting the significance of Snowden's actions.\n\n Score: 5/5\n\nPlease note that the scoring system is subjective and might vary based on individual interpretations. The provided response effectively addresses the user's question and provides a clear, well-structured account of the events surrounding Edward Snowden's disclosure of classified documents.

Figure 9: An example of EFT data. The texts with the colors of green, red and blue are the instruction, response and the LLM-as-a-judge results (explanation and score), respectively

| SFT | SFT + DPO | SFT🔥 + DPO🔥 |
|---|---|---|

**What are the names of some famous actors that started their careers on Broadway?**

| SFT | SFT + DPO | SFT🔥 + DPO🔥 |
|---|---|---|
| 1. Robert Redford\n2. Meryl Streep\n3. Al Pacino\n4. Denzel Washington\n5. Liza Minnelli\n6. James Earl Jones\n7. Bernadette Peters\n8. Nathan Lane\n9. Patti LuPone\n10. Kevin Kline\n11. Matthew Broderick\n12. Whoopi Goldberg\n13. Hugh Jackman\n14. Idina Menzel\n15. Neil Patrick Harris\n16. Audra McDonald\n17. Kristin Chenoweth\n18. John Lithgow\n19. Bebe Neuwirth\n20. Harvey Fierstein\n21. Chita Rivera\n22. Ben Vereen\n23. Joel Grey\n24. John Cullum\n25. John Rubinstein\n26. John Glover\n27. John C. Reilly\n28. John Turturro\n29. John Mahoney\n30. John Larroquette\n31. John Slattery\n32. John Benjamin Hickey\n33. John Gallagher Jr.\n34. John Leguizamo\n35. John Goodman\n36. John Lithgow\n37. John C. McGinley\n38. **John Cusack**\n39. John C. Reilly\n40. John Turturro\n41. John C. McGinley\n42. **John Cusack**\n43. John C. Reilly\n44. John Turturro\n45. John C. McGinley\n46. **John Cusack**\n47. John C. Reilly\n48. John Turturro\n49. John C. McGinley\n50. **John Cusack**\n51. John C. Reilly\n52. John Turturro\n53. John C. McGinley\n54. **John Cusack**\n55. John C. Reilly\n56. John Turt (cut by maximum generation length) | 1. Robert Redford: Redford began his acting career on Broadway in the late 1950s, appearing in productions such as \"Tall Story\" and \"Sunday in New York.\"\n2. Meryl Streep: Streep made her Broadway debut in 1975 in the play \"Trelawny of the 'Wells.'\" She went on to **win three Tony Awards** for her work on Broadway.\n3. Al Pacino: Pacino made his Broadway debut in 1969 in the play \"Does a Tiger Wear a Necktie?\" He went on to win two Tony Awards for his work on Broadway.\n4. Dustin Hoffman: Hoffman made his Broadway debut in 1961 in the play \"A Cook for Mr. General.\" He went on to win **two Tony Awards** for his work on Broadway.\n5. Denzel Washington: Washington made his Broadway debut in **1982 in the play \"A Soldier's Play.\"** He went on to win **two Tony Awards** for his work on Broadway.\n6. Glenn Close: Close made her Broadway debut in 1974 in the musical \"Love for Love.\" She went on to win three Tony Awards for her work on Broadway.\n7. Kevin Spacey: Spacey made his Broadway debut in 1982 in the play \"Ghosts.\" He went on to win **two Tony Awards** for his work on Broadway.\n8. Whoopi Goldberg: Goldberg made her Broadway debut in 1984 in the play \"**Ma Rainey's Black Bottom.**\" She went on to win a Tony Award for her work on Broadway.\n9. Nathan Lane: Lane made his Broadway debut in 1982 in the play \"Present Laughter.\" He went on to win three Tony Awards for his work on Broadway.\n10. Bernadette Peters: Peters made her Broadway debut in **1968** in the musical \"**George M!**\" She went on to win **three Tony Awards** for his work on Broadway.\n11. Patti LuPone: LuPone made her Broadway debut in **1972** in the musical \"**The Threepenny Opera.**\" She went on to win two Tony Awards for her work on Broadway.\n12. L (cut by maximum generation length) | There are many famous actors who started their careers on Broadway. Here are some of the most well-known:\n1. Meryl Streep: Streep made her Broadway debut in 1975 in the play "Trelawny of the 'Wells.'" She went on to win three Academy Awards and become one of the most acclaimed actresses of her generation.\n2. Robert De Niro: De Niro made his Broadway debut in **1965** in the play "**The Indian Wants the Bronx.**" He went on to become one of the most iconic actors in Hollywood, winning two Academy Awards.\n3. Al Pacino: Pacino made his Broadway debut in 1969 in the play "Does a Tiger Wear a Necktie?" He went on to become one of the most respected actors in Hollywood, winning an Academy Award and numerous other accolades.\n4. Dustin Hoffman: Hoffman made his Broadway debut in 1961 in the play "A Cook for Mr. General." He went on to become one of the most versatile actors in Hollywood, winning two Academy Awards.\n5. Denzel Washington: Washington made his Broadway debut in **1982 in the play "A Soldier's Play."** He went on to become one of the most acclaimed actors of his generation, winning two Academy Awards.\n6. Bernadette Peters: Peters made her Broadway debut in **1968** in the musical "**George M!**" She went on to become one of the most beloved performers on Broadway, winning **three Tony Awards**.\n7. Patti LuPone: LuPone made her Broadway debut in **1972** in the musical "**The Threepenny Opera.**" She went on to become one of the most respected performers on Broadway, winning three Tony Awards.\n8. Nathan Lane: Lane made his Broadway debut in 1982 in the play "Present Laughter." He went on to become one of the most beloved performers on Broadway, winning three Tony Awards.\n9. Audra McDonald: McDonald made her Broadway debut in 1992 in the musical "The Secret Garden." She went on to become one of the most acclaimed performers on Broadway, winning six Tony Awards.\n10. Hugh Jackman: Jackman made (cut by maximum generation length) |

**Tell me the bio of Marianne McAndrew (very rare entity)**

| SFT | SFT + DPO | SFT🔥 + DPO🔥 |
|---|---|---|
| 1943-2006\n\nMarianne McAndrew was an American actress best known for her role as Irene Molloy in the **1964** film version of Hello, Dolly! She was born in **Fort Ord, California**, in **1943** and began her acting career in the early 1960s. She appeared in several television shows and films throughout the decade, including **The Fugitive, The Virginian**, and **The Wild Wild West**. In **1964**, she was cast as Irene Molloy in Hello, Dolly!, which earned her a Golden Globe nomination for Best Supporting Actress. After Hello, Dolly!, McAndrew continued to act in films and television shows throughout the 1970s and 1980s. She also appeared on Broadway in the musicals **The Music Man and Mame. McAndrew retired from acting in the early 1990s and passed away in 2006 at the age of 63.** | 1. Marianne McAndrew was an American actress best known for her role as Irene Molloy in the **1968** film version of \"Hello, Dolly!\"\n2. She was born on **July 20, 1943, in Chicago, Illinois**, and began her acting career in the early 1960s.\n3. McAndrew made her film debut in **1965** in the movie \"**The Slender Thread**\" and went on to appear in several other films throughout the 1960s and 1970s, including \"**The Thomas Crown Affair**\" and \"**The Great Gatsby**.\"\n4. In addition to her film work, McAndrew also appeared on television, including guest roles on popular shows such as \"**The Fugitive**\" and \"**Mission: Impossible.**\"\n5. **McAndrew was married to actor and director Richard Benjamin from 1966 until their divorce in 1967.**\n6. **She later married actor and producer Michael Tolan in 1970, and the couple had two children together.**\n7. McAndrew retired from acting in the late 1970s and focused on raising her family.\n8. **She passed away on February 26, 2006, at the age of 62.**\n9. McAndrew's legacy as an actress lives on through her memorable performances in films such as \"Hello, Dolly!\" and \"**The Thomas Crown Affair.**\"\n10. She is remembered as a talented and versatile actress who made a lasting impact on the film industry. | Marianne McAndrew was an American actress who was best known for her role as Irene Molloy in the **1964** film version of the musical "Hello, Dolly!" She was born on **May 27, 1943, in Chicago, Illinois**, and began her acting career in the early 1960s. McAndrew made her film debut in **1963 with a small role in the film "The Cardinal."** She then went on to appear in several other films, including **"The World of Henry Orient" (1964)** and **"The Great Race" (1965)**. However, it was her role as Irene Molloy in "Hello, Dolly!" that brought her the most recognition. In addition to her film work, McAndrew also appeared on television, including in episodes of **"The Virginian"** and "The F.B.I." **She retired from acting in the early 1970s and later worked as a real estate agent in California. McAndrew was married twice and had two children. McAndrew passed away on February 26, 2006, at the age of 62.** Overall, Marianne McAndrew was a talented actress who made a lasting impact on the film industry with her memorable performance in "Hello, Dolly!" |

**Tell me the bio of Ji Sung (very frequent entity)**

| SFT | SFT + DPO | SFT🔥 + DPO🔥 |
|---|---|---|
| Ji Sung is a South Korean actor who was born on February 27, 1977, in Seoul, South Korea. He began his acting career in 1999 and has since become one of the most popular and acclaimed actors in South Korea.\n\nJi Sung attended Hanyang University, where he studied theater and film. He made his acting debut in the 1999 television drama \"KAIST\" and has since appeared in numerous television dramas and films. Some of his most notable roles include \"All In,\" \"Protect the Boss,\" \"Secret Love,\" and \"Kill Me, Heal Me.\"\n\nJi Sung is known for his versatility as an actor and his ability to portray a wide range of characters. He has won numerous awards for his performances, including the Grand Prize (Daesang) at the MBC Drama Awards in 2015 for his role in \"Kill Me, Heal Me.\"\n\nIn addition to his acting career, Ji Sung is also known for his philanthropic work. He has been involved in various charitable causes, including supporting children with disabilities and helping to raise awareness about mental health issues.\n\nJi Sung is married to actress Lee Bo-young, and the couple has a daughter together. He is known for being a devoted family man and often speaks about the importance of spending time with his loved ones.\n\nOverall, Ji Sung is a talented and respected actor who has made a significant impact on the South Korean entertainment industry. | 1. Ji Sung is a South Korean actor who was born on February **29**, 1977, in Seoul, South Korea.\n2. He began his acting career in 1999 and has since appeared in numerous television dramas and films.\n3. Ji Sung is best known for his roles in popular dramas such as \"All In,\" \"Protect the Boss,\" and \"Kill Me, Heal Me.\"\n4. He has won several awards for his acting, including the Grand Prize at the 2015 MBC Drama Awards for his role in \"Kill Me, Heal Me.\"\n5. In addition to his acting career, Ji Sung is also known for his philanthropic work. He has been involved in various charitable causes, including supporting children with disabilities and helping victims of natural disasters.\n6. Ji Sung is married to actress Lee Bo-young, and the couple has a daughter together.\n7. He is known for his versatility as an actor, having played a wide range of characters in his career, from romantic leads to villains.\n8. Ji Sung is also known for his dedication to his craft, often logging long hours on set and undergoing extensive preparation for his roles.\n9. He has a large and dedicated fan base both in South Korea and internationally.\n10. Ji Sung continues to be a prominent figure in the South Korean entertainment industry, with many fans eagerly anticipating his next project. | Ji Sung is a South Korean actor who was born on February **29**, 1977, in Seoul, South Korea. He began his acting career in 1999 and has since become one of the most popular and acclaimed actors in South Korea. Ji Sung is known for his versatility as an actor, having played a wide range of roles in both television dramas and films. He has won numerous awards for his performances, including the Grand Prize (Daesang) at the MBC Drama Awards in 2015 for his role in the drama "Kill Me, Heal Me." Some of Ji Sung's most notable roles include his portrayal of a man with multiple personalities in "Kill Me, Heal Me," a **lawyer** in "Defendant," and a doctor in "Doctor John." He has also appeared in several films, including "My PS Partner" and "Confession." In addition to his acting career, Ji Sung is also known for his philanthropic work. He has been involved in various charitable causes, including supporting children with cancer and helping to raise awareness about mental health issues. Overall, Ji Sung is a talented and respected actor who has made a significant impact on the South Korean entertainment industry. |

Figure 10: Generation comparisons for instructions from Alpaca Eval and Biography (very rare and frequent entities). Determined through manual verification using Google search, red denotes incorrect identified facts while pink indicates unverified facts; e.g., we cannot search relevant pages about Ji Sung's involvement in charitable causes but also cannot dismiss the possibility of his contributions. Note that the popularity of an entity is defined by its occurrence and page views in Wikipedia, which are provided by Min et al. [2023].

