# OpenReview forum: "FLAME : Factuality-Aware Alignment for Large Language Models"
_NeurIPS.cc/2024/Conference — NeurIPS 2024 poster_

### Official Review · Reviewer_b1G9 · 2024-07-07

**Soundness:** 3
**Presentation:** 3
**Contribution:** 3
**Rating:** 6
**Confidence:** 5

**Summary:**

This work studies how to do alignment for large language models to improve their factuality. The focus of this work is on SFT and DPO. The motivation behind this work is a pilot study which shows more factual data does not always lead to a more factual model. To resolve this issue, the proposed Flame framework (1) handles fact-based and non-fact-based examples differently; (2) uses few-shot generated examples from the model itself for fact-based SFT; (3) builds a reward model specifically for factuality (via atomic fact decomposition, retrieval augmented claim verification, etc.) Experiments on multiple datasets demonstrate that Flame can improve the model's factuality without hurting other capabilities (e.g., instruction following). Ablations are also conducted to measure the gain from each individual step.

**Strengths:**

1. The motivation is clear and reasonable. I like using a simple and quick pilot experiment to demonstrate the main motivation of this paper.

2. The idea is straightforward and effective. The high level framework can applied to many different systems.

3. Ablation experiments are conducted to show the gain from each step. The effectiveness for both SFT and DPO are clear.

**Weaknesses:**

1. No external baselines are used in the comparison. It would be great to compare the flame model with other related approaches (e.g., few-shot prompting, sampling multiple responses, and reranking using FactScore or the reward model). I know these approaches are not directly comparable, however, it will still be valuable to understand the relative trends, especially since approaches such as few-shot prompting are used in data generation.

2.  It will be great to conduct human evaluations even just on a few examples.

3. The whole pipeline involves a number of components. While many details are presented in the appendix, low-level details like few-shot prompts, and implementation of fact decomposition are omitted. Adding these details will be super valuable for future work to build similar systems. It would be even better if the authors decide to release the code.

**Questions:**

1. In the pilot experiment, doing DPO with FS seems to work reasonably well. Have you tried similar approaches in the real experiments?

**Limitations:**

Limitations are discussed in Sec. A.6 in the appendix.

---

> ### Author Rebuttal · Authors · 2024-08-06
>
> 1. Re: No external baselines are used in the comparison.
>
> We thank you for the suggestion. As far as we know, the existing best approach to factual alignment is the method introduced by Tian et.al. [1]. Note that, this approach conducts factual alignment directly on the target task (e.g., biography generation) while we conduct factual alignment on more diverse and general instruction following tasks. Although it is hard to conduct a direct comparison with the baseline, as mentioned in line 277, we do provide a baseline which mirrors the approach by Tian et.al. [1], $\mathrm{SFT }+ \mathrm{DPO}^{\textrm{fact}}$ as shown in row 3 of Table 3, where factuality is the only objective of DPO. The main purpose of our paper is not to propose a state-of-the-art approach for factual alignment. Instead, we find the side effect of applying factual alignment to instruction tuning and manage to mitigate the side effect.
>
> 2. Re: It will be great to conduct human evaluations even just on a few examples.
>
> Thanks for the suggestions. We agree that adding human evaluations would make the result more convincing. We have conducted manual evaluations for two baselines and FLAME on a few representative cases from Alpaca Eval and Biography in case study, as shown in Appendix Figure 10. We will make it clear in the section of case study.
>
> 3. Re: Low-level details like few-shot prompts, and implementation of fact decomposition are omitted.
>
> Thanks for the suggestions. Below are the low-level details about our few-shot prompts and fact decomposition, which we will add into our revised manuscript.
>
> For the detail of few-shot prompts, as mentioned in lines 195--198, for each instruction $x$, we retrieve the top-5 similar instructions ($x_0 \cdots x_4$) along with the corresponding human responses ($\mathrm{Human}(x_0) \cdots \mathrm{Human}(x_4)$) from OASST (our seed data with 3.2K instruction--human response pairs). Then, concatenating them as the few-shot prompt $x_4, \mathrm{Human}(x_4), x_3, \mathrm{Human}(x_3), \cdots, x_0, \mathrm{Human}(x_0), x$ to elicit the response from the pre-trained LLM.
>
> For the implementation of fact decomposition, we reuse the code from FActScore to conduct fact decomposition but replace the GPT3.5 with our fine-tuned Llama2 7B model on the public datasets (i.e., RoSE benchmark [2], CLAIMDECOMP [3] and EXPERT QA dataset [4]). Note that the purpose of fine-tuning our Llama2 7B for atomic fact decomposition is to save the cost of creating factuality preference training data.
>
> 4. Re: In the pilot experiment, doing DPO with FS seems to work reasonably well. Have you tried similar approaches in the real experiments?
>
> DPO with FS can be considered the approach by Tian et.al. [1] and as mentioned in point 1, in our main experiment for instruction tuning, we do provide a baseline as shown in row 3 of Table 3, which is mentioned in line 277. We will make the statement more clear in the revised version.
>
> [1] Fine-tuning Language Models for Factuality. https://arxiv.org/abs/2311.08401
>
> [2] https://github.com/Yale-LILY/ROSE
>
> [3] https://github.com/jifan-chen/subquestions-for-fact-checking
>
> [4] https://github.com/chaitanyamalaviya/ExpertQA

---

> > ### Comment · Reviewer_b1G9 · 2024-08-12
> >
> > Thank you for the response! Adding these clarifications and additional details in the revised version will be very helpful. I will keep my original positive rating of 6.

---

> > > ### Author Response · Authors · 2024-08-14
> > >
> > > Thank you for the helpful suggestions. We will revise our manuscript accordingly.

---

### Official Review · Reviewer_6wxd · 2024-07-10

**Soundness:** 3
**Presentation:** 4
**Contribution:** 2
**Rating:** 5
**Confidence:** 4

**Summary:**

This paper shows that training on new or unfamiliar knowledge can promote hallucination and that reward functions in standard RL often inadequately capture factuality. The authors propose a factuality-aware alignment method that first identifies instructions as fact-based or non-fact-based. For fact-based instructions, they employ adapted techniques in respective SFT and RL to generate additional training data, thereby reducing the hallucination of the model's responses.

**Strengths:**

* The paper conducts a pilot study that highlights the limitations of SFT and RL in capturing factual knowledge. This study provides valuable insights into data selection for LLM alignment training.
* The proposed dual-stage factuality-aware method improves factuality without compromising the instruction-following capabilities for both SFT and RL stages.

**Weaknesses:**

* The proposed strategy to create SFT and DPO training data using the generated responses from the LLM itself is limited to the knowledge learned within the original model. This approach may struggle with instructions that the original model cannot generate factual answers for.
* The proposed strategy relies on accurately identifying the instruction type initially, which is limited by the model's ability to correctly classify the instruction type.
* In the pilot study, it is unclear whether the $PT$ and $PT^{RAG}$ are evaluated using the same protocol as other methods. If they are, the FS score decreases after both SFT and DPO, which contradicts the claim that "fine-tuning LLMs on their own generations appears to be crucial for factual alignment."
* While the results in Table 2 and 3 indicate that eliciting knowledge from the model itself can enhance factuality compared to introducing more factual but unknown knowledge, it does not improve the FS of the $PT$, which achieves a score of 53.1 on the Biography task with just 5-shot demonstrations.
* As discussed in Sec 5.5, conducting fact checks and computing factuality rewards solely for fact-based sentences can lead to more factuality errors. Clarification is needed on how FS is calculated for the experiments in Sec 5.2 and 5.3.

**Questions:**

Please refer to the Weaknesses.

**Limitations:**

Yes, the authors have discussed the limitations of the metric for evaluating factuality.

---

> ### Author Rebuttal · Authors · 2024-08-06
>
> 1. Re: This approach may struggle with instructions that the original model cannot generate factual answers for.
>
> We thank you for the insight. This is exactly what we found in the paper; that is, it is challenging to teach LLMs to learn new knowledge in the finetuning stage. Inevitably, forcing LLMs to learn new knowledge (e.g., learning from RAG or human written response) may cause more hallucination, which is also found in the concurrent paper [1]; thus, we propose to fine-tune LLMs only on its own generated responses. We agree that it is worth conducting exploring how to inject new knowledge to LLMs during fine-tuning stage without any side effects but this is orthogonal to our work. We will make our claim more clear in our revised manuscript.
>
> 2. Re: The proposed strategy relies on accurately identifying the instruction type initially, which is limited by the model's ability to correctly classify the instruction type.
>
> Yes, we admit that FLAME does require a module to distinguish the types of instructions. However, instead of treating it as a weakness of our approach, we believe that it is the important finding of our paper. That is, not all instructions require factual responses, and overly optimizing for factuality could lead to degradation in instruction following capability (condition 3 vs 4 in Table 3). Furthermore, the previous study [2] also find that different instructions require different evaluation strategies. Finally, we also demonstrate that the task of classifying the instruction type can be easily done by an instruction following fine-tuned model rather than a specially fine-tuned model.
>
> 3. Re: In the pilot study, it is unclear whether the $\mathrm{PT}$ and $\mathrm{PT}^{\text{\tiny RAG}}$ are evaluated using the same protocol as other methods.
>
> Yes, in our pilot study $\mathrm{PT}$ and $\mathrm{PT}^{\text{\tiny RAG}}$ are evaluated using the same protocol as other methods. However, this does not mean the results contradict our claim: fine-tuning LLMs on their own generations appears to be crucial for factual alignment. First of all, the responses from $\mathrm{PT}$ and $\mathrm{PT}^{\text{\tiny RAG}}$ are generated using 5-shot demonstrations while the responses from SFT and DPO models (only fine-tuned on 495 instructions for biography generation) are generated with 0 shot. Especially, $\mathrm{PT}^{\text{\tiny RAG}}$ also concatenate retrieved evidence; thus, it is reasonable that $\mathrm{PT}^{\text{\tiny RAG}}$ shows far better factual accuracy. Although SFT (fine-tuned with a few examples) zero-shot generation shows a slight factuality degradation compared to $\mathrm{PT}$ 5-shot prompts, we do see that the DPO model improves over the $\mathrm{PT}$ model in Table 1 (condition 5 vs 1). To be clear, our claim comes from the observation that SFT and DPO fine-tuned with LLM’s own generation versus RAG generation (3 vs 4, 5 vs 6) even though RAG generation is shown to be far more factual.
>
> 4. Re: SFT and DPO do not improve models’ factuality over 5-shot prompts.
>
> We want to clarify that in our paper, improving LLMs’ factuality does not mean that we can make the instruction following fine-tuned LLMs generate more factual responses than the pre-trained LLMs with few-shot demonstrations. In fact, there are previous papers which find that instruction tuning or alignment may degrade LLMs’ performance on the standard knowledge benchmarks, called alignment tax [3]. The alignment tax can explain why our DPO fine-tuned model (only on biography generation) can improve over $\mathrm{PT}$ (Table1 condition 5 vs 1) but the DPO fine-tuned model (on diverse instruction following tasks) cannot improve over $\mathrm{PT}$ (condition 7 in Table 3 vs condition 0 in Table 2). Again, we want to highlight that the main purpose of the paper is to study why instruction following fine-tuning tends to cause hallucination and how to mitigate the issue rather than addressing the alignment tax. Thanks for the insightful question, we will add further explanation in our revised paper.
>
> 5. Re: As discussed in Sec 5.5, conducting fact checks and computing factuality rewards solely for fact-based sentences can lead to more factuality errors. Clarification is needed on how FS is calculated for the experiments in Sec 5.2 and 5.3.
>
> All our FActScore evaluation is based on the assumption that all the sentences in the responses are fact-based ones for two reasons. First, as pointed out in our ablation study, excluding the non-fact-based sentences may cause noise. Second, from our eyebrow check, LLMs’ generation for Biography, Alpaca Fact and FAVA datasets are mostly fact-based sentences. Figure 10 in Appendix showcases a few LLMs’ responses for Biography and Alpaca Fact.
>
> [1] Does Fine-Tuning LLMs on New Knowledge Encourage Hallucinations? https://arxiv.org/abs/2405.05904
>
> [2] Branch-Solve-Merge Improves Large Language Model Evaluation and Generation. https://arxiv.org/abs/2310.15123
>
> [3] Training language models to follow instructions with human feedback https://arxiv.org/pdf/2203.02155

---

> > ### Comment · Reviewer_6wxd · 2024-08-11
> > **Reply**
> >
> > I thank the authors for their insightful replies to my review. They are indeed helpful. I still have some concerns regarding your evaluation assumption that all the sentences in the responses are fact-based, which might not be entirely fair. The baseline model fine-tuned on high-quality instruction data, as pointed out by [1], might generate more stylistic tokens, such as "Of course." Could fact-checking these sentences decrease the FactScore? The evaluation deserves more rigorous and comprehensive discussion. (Note that Reviewer HVe5 also pointed out the limitation of your evaluation protocol.)
> >
> > Given most of my concerns have been addressed, I would like to raise my score from 4 to 5.
> >
> > [1] [The Unlocking Spell on Base LLMs: Rethinking Alignment via In-Context Learning](https://arxiv.org/abs/2312.01552)

---

> > > ### Author Response · Authors · 2024-08-11
> > >
> > > We thank you all your helpful feedback to improve our manuscript. Regarding to your last concern, we thank you for the reference and this observation indeed inspires us to conduct the experiment to filter out non-fact-based sentences in the ablation study. As for evaluation, we have conduct a manual check on those responses for Bio, Alpaca-Fact, and Fava. For Bio and Fava, we observe almost all the generated sentences are fact-based ones. For Bio and Alpaca-Fact, we have some examples in Table 10 and for Fava, here we give two examples from SFT. Our observation is that for those fact-based instructions, LLMs' responses consist mostly of fact-bases sentences. We will revise our manuscript and make our assumption based on the observation for evaluation clear.
> > >
> > > **Case1**
> > > What are the Golden Lion movies at the Venice Film Festival whose production countries are in Asia?
> > >
> > > 1. The 2018 film "Shadow" by Chinese director Zhang Yimou.2. The 2019 film "The Wild Goose Lake" by Chinese director Diao Yinan.3. The 2020 film "Nomadland" by American director Chloé Zhao, which was partially shot in South Korea.4. The 2021 film "The Hand of God" by Italian director Paolo Sorrentino, which was partially shot in Thailand.5. The 2022 film "All the Beauty and the Bloodshed" by American director Laura Poitras, which was partially shot in Japan.
> > >
> > > **Case2**
> > > Why is white phosphorous so dangerous?
> > >
> > > White phosphorus is dangerous because it is highly flammable and can spontaneously ignite in air, releasing toxic fumes. It is also highly reactive and can cause severe burns and tissue damage on contact with skin. In addition, white phosphorus is a strong oxidizing agent and can react violently with other substances, such as organic materials, to cause fires or explosions.

---

### Official Review · Reviewer_XBcd · 2024-07-12

**Soundness:** 3
**Presentation:** 3
**Contribution:** 2
**Rating:** 4
**Confidence:** 4

**Summary:**

This paper addresses the issue of factual inaccuracy, or "hallucination," in Large Language Models (LLMs). The authors identify factors that lead to the generation of false facts during supervised fine-tuning (SFT) and reinforcement learning (RL). They propose FLAME, a novel alignment method that incorporates factuality-aware SFT and direct preference optimization (DPO) to guide LLMs towards more factual responses without compromising their ability to follow instructions. Experiments demonstrate FLAME's effectiveness in enhancing factuality while maintaining helpfulness.

**Strengths:**

1. The ablation experiments provides comprehensive insights into the effectiveness of DPO and SFT in mitigating hallucination.
2. The method proposed in this paper attempts to balance instruction following and factuality. It relies on model self-construction data, and does not depend on external proprietary models.

**Weaknesses:**

1. The baselines compared in this work are limited to different settings of SFT and DPO only. The baselines in the paper should at least include the work [1]. This prior work also uses DPO and algorithms, and the only difference seems to be data construction. The paper should compare with this work to demonstrate that its algorithm truly achieves a balance between instruction following and factuality.
2. In addition to the works listed in the related work, there are some works whose methods are somewhat similar to this paper, such as [2] [3], etc. The paper may need to add explanations of the differences between these methods to clarify its own novelty.

[1] Fine-tuning Language Models for Factuality. https://arxiv.org/abs/2311.08401

[2] Self-Alignment for Factuality: Mitigating Hallucinations in LLMs via Self-Evaluation. https://arxiv.org/abs/2402.09267

[3] GRATH: Gradual Self-Truthifying for Large Language Models. https://arxiv.org/pdf/2401.12292

**Questions:**

1. In comparison to training, have the authors considered comparing representation editing baseline methods?
2. Could the authors supplement experiments on the TruthfulQA-MC in Table 4 to provide a measure of multi-choice performance?

**Limitations:**

The authors have addressed some limitations of their work in the Appendix, which is commendable.

---

> ### Author Rebuttal · Authors · 2024-08-06
>
> 1. Re: The baselines compared in this work are limited to different settings of SFT and DPO only.
>
> We thank your suggestion to compare with the baseline from Tian et.al. [1]. First of all, we want to clarify that Tian et.al. [1] mainly focus on fine-tuning LLMs on a specific task (e.g. biography generation) while FLAME focuses on more general instruction following tasks, where biography generation is a subtask for us to evaluate factuality. Thus, it is hard to make a fair comparison between the two approaches. Nevertheless, as we mention in Section 5.3 (line 277), we do apply the approach from Tian et.al. [1] to our alignment training as $\mathrm{SFT}^{\textrm{fact}}$ and $\mathrm{DPO}^{\textrm{fact}}$, where factuality is the only optimization objective. This result (row 4 vs 3 in Table 3) supports our claim from related work (lines 84--85) that solely focusing on factual alignment may impact LLMs’ instruction following capability. Furthermore, we also point out the importance of finding those fact-based instructions among the instructions, which is ignored by Tian et.al. [1] if we only conduct factual alignment on biography generation tasks. In other words, FLAME is the extension of Tian et.al. [1] to more general instruction tuning tasks. We will revise the manuscript to clarify that we have done our best to compare with the existing factuality fine-tuning method [1] in our main experiment.
>
> 2. Re: The paper may need to add explanations of the differences between these methods to clarify its own novelty.
>
> We thank you for the good references and do agree that we need to compare the more recent work in our related work [2][3]. The main contribution of the related work[2][3] focuses on using the LLM itself as the factuality judge to enhance its factuality. Similar to Tian et.al. [1], they mainly focus on fact-based instructions and can be considered as an improved version of FactTuneMC from Tian et.al. [1]; thus, ignoring the impact of factuality alignment on LLMs’ instruction following capability. Furthermore, the proposed factuality alignment approaches from [2][3] can be integrated into FLAME to create more accurate factuality pairs. We will add the comparison and cite the references in our revised version.
>
>
> 3. Re: In comparison to training, have the authors considered comparing representation editing baseline methods?
>
> We thank you for the suggestions. Although representation editing baseline approaches (e.g., DoLa [4] and ITI [5]) also show promising results, these approaches rely on a development set of the target task to tune the hyperparameters and focus on a single evaluation metric. As we have shown in our main experiment, focusing only on improving factuality may sacrifice models’ instruction following capability. Furthermore, as shown in Tian et.al. [1], the representation editing approaches are not more effective than fine-tuning for factuality, which is one of our baseline approaches (row 3 in Table 3).
>
> 4. Re: Could the authors supplement experiments on the TruthfulQA-MC in Table 4 to provide a measure of multi-choice performance?
>
> We thank you for the suggestion. Below is the comparison of TruthfulQA-MC (zero-shot). Note that, we do not observe significant differences between models on TruthfulQA-MC task. This is possibly because we mainly focus on the tasks of long-form response generation while TruthfulQA-MC task is formed by short-form answers. The discrepancy between improving LLMs’ factuality on long-form and short-form generation is also found by the previous work [4]. We will add the experiment and explanation to our revised manuscript.
>
> | TruthfulQA_MC       | MC1 | MC2 | MC3 |
> | ------------ | ----------- | ------ | ------ |
> |  (0) Chat   | 32.2| 50.2 |25.4|
> |  (1) $\mathrm{SFT}$  |30.8| 45.7| 23.9|
> |  (2) + $\mathrm{DPO}$|30.5| 46.0| 23.4|
> |  (3) + $\mathrm{DPO}^{\textrm{fact}}$|31.8|46.8|24.3|
> |  (4) + $\mathrm{DPO}^{\textrm{FLAME}}$| 30.8| 46.0|23.6|
> |  (5) $\mathrm{SFT}^{\textrm{FLAME}}$  |29.9|44.8|22.5|
> |  (6) + $\mathrm{DPO}$  |31.5|47.0|24.0|
> |  (7) + $\mathrm{DPO}^{\textrm{FLAME}}$| 30.5| 45.4| 23.1|
>
> Reference:
>
> [1] Fine-tuning Language Models for Factuality. https://arxiv.org/abs/2311.08401
>
> [2] Self-Alignment for Factuality: Mitigating Hallucinations in LLMs via Self-Evaluation. https://arxiv.org/abs/2402.09267
>
> [3] GRATH: Gradual Self-Truthifying for Large Language Models. https://arxiv.org/pdf/2401.12292
>
> [4] DoLa: Decoding by Contrasting Layers Improves Factuality in Large Language Models. https://arxiv.org/abs/2309.03883

---

> > ### Comment · Reviewer_XBcd · 2024-08-13
> > **Reply to the authors**
> >
> > Thank you for your response regarding the additional results. I've read the response in detail and decided to keep my score and judgments.

---

> > > ### Author Response · Authors · 2024-08-14
> > >
> > > We thank you for the discussion and will update the suggested experiments and explanation in our revised manuscript.

---

### Official Review · Reviewer_HVe5 · 2024-07-14

**Soundness:** 3
**Presentation:** 3
**Contribution:** 3
**Rating:** 6
**Confidence:** 3

**Summary:**

The paper discusses a novel alignment method to enhance the factual accuracy of LLMs. The authors observe that conventional alignment processes, which include SFT and RL, often result in the generation of false facts or 'hallucinations'. To address this, they introduce factuality-aware alignment (FLAME), which includes factuality-aware SFT and RL through direct preference optimization. FLAME identifies factors leading to hallucination and adapts the training process to reduce the generation of false claims. Experiments demonstrate that FLAME guides LLMs to produce more factual responses without compromising their ability to follow instructions. The paper contributes to the field by tackling the issue of maintaining helpfulness while improving the factuality of AI-generated content.

**Strengths:**

- Clear and Logical Structure: This paper is well-organized and presents its findings with a logical flow, making it easy to follow.
- In-depth Analysis of Hallucination: The paper thoroughly analyzes the factors contributing to hallucination during the SFT and RL phases of language model alignment. It identifies key issues: training on unfamiliar data can reduce factual accuracy, and standard RL reward functions often prioritize longer, more detailed responses, potentially encouraging the model to fabricate information.
- Innovative Solution:  The proposed FLAME is a novel alignment approach that effectively addresses hallucination without compromising the model's ability to follow instructions. By extending both SFT and RL, FLAME tackles a critical issue in LLMs, ensuring more accurate and reliable information generation.
- Comprehensive Evaluation: The paper thoroughly evaluates FLAME's effectiveness in improving both factuality and instruction-following abilities. Experiments demonstrate that models aligned using FLAME achieve significantly higher FactScore compared to standard alignment methods, without sacrificing their helpfulness.

**Weaknesses:**

This paper is well-written and makes a valuable contribution to the LLM alignments. I only have several minor concerns as follows:
- Model Size and Generalizability: The paper focuses solely on the LLaMA2-70B model. It would be beneficial to investigate whether FLAME's effectiveness extends to smaller models, such as 7B or even smaller, given that the factuality-aware SFT relies on self-supervision through few-shot prompting.
- Evaluation Metrics and Human Assessment: While FactScore is a valuable metric, it has limitations. It assumes Wikipedia as the definitive source of truth and may not be suitable for broader domains. Using a more comprehensive metric like Veriscore [1] could provide a more nuanced evaluation (I understand that Veriscore is a recently released method, so this is a suggestion for the future version of this paper). Additionally, incorporating human evaluation would strengthen the analysis. A manual assessment of factuality and helpfulness would provide valuable insights and increase the persuasiveness of the findings.
- Multi-faceted Evaluation: The paper primarily focuses on instruction following and factuality. However, other crucial aspects of LLM capabilities, including knowledge, reasoning, and code generation, should also be considered. It would be insightful to evaluate the performance of FLAME-trained models on standard benchmarks like MMLU, GSM8K, and HumanEval to assess potential trade-offs in these areas.

**Questions:**

- While FLAME primarily focuses on DPO, can it also be applied to conventional reinforcement learning from human feedback (RLHF) methods like PPO?
- Are there plans to release the code and models trained using FLAME for the research community to replicate your methods?

**Limitations:**

The authors adequately addressed the limitations and broader impacts.

---

> ### Author Rebuttal · Authors · 2024-08-06
>
> 1. Re: Model Size and Generalizability. It would be beneficial to investigate whether FLAME's effectiveness extends to smaller models, such as 7B or even smaller, given that the factuality-aware SFT relies on self-supervision through few-shot prompting.
>
> We thank you for the helpful suggestion. We believe our finding is also valid for smaller models. In our pilot study, the condition of row 3 in Table 1 is equal to  factuality-aware SFT with self-supervision through few-shot prompting. The pilot study on smaller models motivates us to extend our study to more general instruction tuning rather than focusing on biography generation. However, we have to admit that it is more challenging to fine-tune a smaller model (e.g., Llama2 7B) to follow instructions, especially that we only use 3200 SFT samples from OASST dataset. Not to mention applying the self rewarding [1] methodology to improve the models’ instruction following capability. Note that we choose not to rely on the large size and high quality SFT data (e.g., Alpaca cleaned dataset or UltraFeedback) generated from GPT4.
>
> 2. Re: Factuality Evaluation
>
> Although FActScore is the best automatic factuality evaluation tool we can get access to, we agree and discuss the limitation of FActScore in the Section of Limitation. We thank you for referring to the more recent factuality evaluation tool (e.g., Veriscore).
>
> 3. Re: Multi-faceted Evaluation
>
> We thank you for the helpful suggestion. Below is the comparison between the baseline and factuality-aware fine-tuned models on MMLU and GSM8K. Their results are very close. We will add the evaluation on standard benchmark in the appendix.
>
> |       | MMLU | GSM8K |
> | ------------ | ----------- | ------ |
> |  (2) $\mathrm{SFT} + \mathrm{DPO}$|69.34| 59.28|
> |  (7) $\mathrm{SFT}^{\textrm{FLAME}} + \mathrm{DPO}^{\textrm{FLAME}}$| 69.05| 58.22|
>
> 4. Re: While FLAME primarily focuses on DPO, can it also be applied to conventional reinforcement learning from human feedback (RLHF) methods like PPO?
>
> We thank you for the question. Yes, we believe FLAME can be also applied to PPO. As shown in our ablation study (the third row in Table 6), we can combine the rewards of instruction following capability and factuality into a single scalar reward. With the single scalar reward, we can conduct PPO for preference fine-tuning. However, due to the complexity and instability of fine-tuning with PPO, we instead use DPO as a proof-of-concept in our experiment. We will add this discussion in Section 5.5 in the revised manuscript.
>
> 5. Re: Are there plans to release the code and models trained using FLAME for the research community to replicate your methods?
>
> Thanks for the suggestions. We will consider releasing the code or the scripts for creating the preference training data.

---

> > ### Comment · Reviewer_HVe5 · 2024-08-09
> >
> > Thanks for the response. However, I think most of my concerns have not been addressed (generalizability, performance degradation on other abilities, and uncertainty of code/data release). After reading other reviewers' comments, I would like to lower my rating to weak accept.

---

> > > ### Author Response · Authors · 2024-08-10
> > >
> > > We thank you for the helpful feedback and comment. We will try our best to address your concern in the revised version.
> > >
> > > For performance degradation, we assume the degrade on the benchmark is relatively slight and again since we are focusing on long-form generation and not expect to see performance gain on the benchmark with short-form answers. The discrepancy between improving LLMs’ factuality on long-form and short-form generation is also found by the previous work [1].
> > >
> > > [1] DoLa: Decoding by Contrasting Layers Improves Factuality in Large Language Models. https://arxiv.org/abs/2309.03883

---

> > > > ### Comment · Reviewer_HVe5 · 2024-08-14
> > > >
> > > > Thanks for the response. Given the available information of the current version of the paper and the results provided during the rebuttal. I would like to keep my rating as 6.

---

> > > > > ### Author Response · Authors · 2024-08-14
> > > > >
> > > > > We thank you for the helpful suggestions, especially the model size generalization and potential side effects on other benchmarks. We will revise our manuscript accordingly.

---

### Decision · Program_Chairs · 2024-09-25

**Decision:**

Accept (poster)

**Comment:**

This paper investigates hallucination in Large Language Models (LLMs) and explores methods for making LLM alignment more factual. It offers two main contributions:
* It demonstrates that the relatively standard alignment processes based on supervised fine-tuning (SFT) and reinforcement learning (specifically, DPO) actually encourage LLM hallucinations.
* Based on this finding, the paper proposes a set of factuality-aware methods (FLAME) to improve SFT and reinforcement learning.

Experiments show that FLAME enhances factuality while maintaining the helpfulness of the AI assistant. Unlike prior work (e.g., Tian et al. 2023), which focuses on specific target tasks, this paper's approach applies factual alignment to diverse and general instruction-following tasks.

The discussion with the authors mainly addressed clarity issues and the addition of supplemental results. Two reviewers expressed concerns about the lack of comparison with a baseline, such as Tian et al. 2023. However, a direct comparison is challenging because the paper’s approach is more general and applies to the overall alignment process rather than a specific target task. Nevertheless, the authors provided an ablation study that differentiates the contributions of Tian et al. from those of the current paper (row 3 vs. 4 in Table 3), highlighting the benefits of the paper’s approach.

Most reviewers were ultimately positive about the submission, though they suggested strengthening the work with a more substantial human evaluation. The agreed-upon changes, such as expanding the related work section, will need to be included in the camera-ready version of the paper.

Overall, I agree with the reviewers and recognize the strengths of the paper. While the method presented may seem relatively straightforward, it is well-motivated and effective. The paper also offers valuable analyses and ablations that provide insights into data selection for LLM alignment training. Consequently, it represents a meaningful contribution to the field on the important topic of reducing hallucinations.